# Fast Estimation for Forest Matrix of Signed Graphs

**Haoxin Sun** [1]    **Zhongzhi Zhang** [1]

## Abstract

The forest matrix of a signed graph plays an important role in network science and social opinion dynamics, yet existing algorithms are mainly designed for unsigned graphs and are difficult to extend to signed graphs. In this paper, we study the problem of efficiently estimating the forest matrix of signed graphs with $n$ nodes and introduce the signed forest matrix theorem, which establishes the relationship between generalized spanning converging forests and the forest matrix. Based on this result, we propose a novel algorithm GSCF, built on a variant of loop-erased random walks, to generate generalized spanning converging forests in expected $O(n)$ time. We further develop two sampling algorithms, FMDE and FMDE+, for estimating the diagonal of the forest matrix, both with time complexity $O(ln)$, where $l$ is the number of samples. Extensive experiments on various signed graphs show that our methods achieve high estimation accuracy, significantly improve computational efficiency, and scale to graphs with over twenty million nodes. Our source code is publicly available on https://github.com/HaoxinSun98/SignedForestDiagonal.

## 1. Introduction

The forest matrix, denoted as $Q = (I + L)^{-1}$, where $L$ is the Laplacian matrix, is a powerful tool in network science. Its properties and applications have been studied in extensive studies such as (Chebotarev & Shamis, 1995; 1997; 1998; 2006). In recent years, the scope of applications for the forest matrix and its variants has expanded significantly, influencing fields such as opinion dynamics (Gionis et al., 2013; Sun & Zhang, 2023; Zhou et al., 2024; Xu et al., 2021; Neumann et al., 2024; Sun et al., 2026), graph signal processing (Pilavcı et al., 2021; Pilavci et al., 2020) and

---
[1] College of Computer Science and Artificial Intelligence, Fudan University, Shanghai, China. Correspondence to: Zhongzhi Zhang <zhangzz@fudan.edu.cn>.

*Proceedings of the 43rd International Conference on Machine Learning*, Seoul, South Korea. PMLR 306, 2026. Copyright 2026 by the author(s).

Markov processes (Avena & Gaudillière, 2018; Avena et al., 2018). In particular, the diagonal entries of $Q$ are crucial and have recently been the subject of studies focusing on their efficient computation (Jin et al., 2019; van der Grinten et al., 2021; Sun & Zhang, 2024). The entries of the forest matrix are also pivotal for determining the forest closeness centrality of networks (Jin et al., 2019; van der Grinten et al., 2021) and have been closely associated with determinantal point processes in machine learning (Kulesza et al., 2012). Additionally, they provide valuable insights through electrical interpretations in multi-agent and network-based problems (Rossi et al., 2017).

With the growing recognition of competitive interactions in real systems, signed graphs have attracted significant scholarly attention (Halo et al., 2024; Sun et al., 2022; 2020; Xiao et al., 2020; Tzeng et al., 2020; Singh & Adhikari, 2017; Friedkin & Johnsen, 1990; Carrasco & Sun, 2025). The introduction of negative edges modifies the properties and computational challenges related to the forest matrix in these graphs. For instance, the forest matrix in signed graphs is no longer row-stochastic, and the conventional forest matrix theorem (Chebotarev & Shamis, 2006; 1998), which links the forest matrix to spanning forests, no longer applies. The forest matrix is central to the signed Friedkin-Johnsen (FJ) model, an influential model in opinion dynamics that addresses both cooperative and antagonistic relationships, offering a nuanced view of human relational dynamics (Xu et al., 2020; Rahaman & Hosein, 2021; He et al., 2020; 2022; Tang et al., 2016; Halo et al., 2024). Particularly, the diagonal elements of the forest matrix in the signed FJ model determine the weight each agent assigns to their initial opinions at equilibrium, with great significance in node ranking and centrality measures. In addition, the forest matrix elements are also closely related to the expressed opinions of the individuals in the signed FJ model, which is the basis for the study of opinion dynamics. However, existing algorithms (Jin et al., 2019; van der Grinten et al., 2021; Sun & Zhang, 2024) fail to effectively estimate the elements of the forest matrix in signed graphs due to these altered properties. Specifically, the methods proposed in (Jin et al., 2019; van der Grinten et al., 2021) rely on fast Laplacian solvers (Cohen et al., 2014), which are not applicable to signed graphs. Additionally, the sampling method developed in (Sun & Zhang, 2024) fails to run, as it relies on the

forest matrix theorem for unsigned graphs, which does not hold in the signed case. Consequently, a theoretically guaranteed estimation algorithm for approximating the elements of the forest matrix of signed graphs is imperative.

In this paper, we delve deeply into the problem of efficiently computing the forest matrix in signed digraphs with $n$ nodes, aiming to address the challenges and limitations of existing algorithms. The primary contributions of this work are summarized as follows:

(i) We introduce a new forest matrix theorem specifically tailored for signed graphs. This theorem establishes the foundational relationship between generalized spanning converging forests and the forest matrix, and elucidates several key properties of the forest matrix in the context of signed graphs.

(ii) To generate a generalized spanning converging forest, we propose a novel algorithm, denoted as GSCF. This algorithm is based on a variant of the loop-erased random walk, and we demonstrate that its expected running time is $O(n)$, making it highly efficient for practical applications.

(iii) We develop two rapid sampling algorithms, FMDE and FMDE+, designed to estimate the diagonal of the forest matrix. Both algorithms operate with a time complexity of $O(ln)$, where $l$ is the number of samples. FMDE+, an enhancement over FMDE, incorporates additional information that results in superior theoretical and experimental performance. We also develop an algorithm FJOE, to estimate the expressed opinion of the signed FJ model as an application of our proposed methods.

(iv) Through extensive experiments conducted on various signed graphs, we demonstrate that our algorithms not only achieve high estimation accuracy but also significantly enhance computational efficiency. Additionally, our approaches are scalable to massive graphs, effectively handling networks with more than twenty million nodes.

## 2. Related Work

The forest matrix is closely related to spanning rooted forests in graphs, as established by the forest matrix theorem (Chebotarev & Shamis, 2006; 1997; 1998). Recent research has increasingly focused on computing quantities or solving optimization problems associated with the forest matrix and its variants. For instance, efforts have been made to compute the PageRank vector (Liao et al., 2023; 2022), solve linear systems in graph signal processing (Pilavcı et al., 2021; Pilavcı et al., 2020), address optimization problems in opinion dynamics (Sun & Zhang, 2023), and estimate the trace of the forest matrix (Pilavcı et al., 2022; Pilavcı et al., 2022). The algorithms developed for these problems are predominantly sampling-based, relying on the theoretical

foundation of the forest matrix theorem and utilizing variants of Wilson's algorithm for loop-erased random walks to sample spanning trees or forests (Wilson, 1996).

Efficient computation of the diagonal elements of the forest matrix has recently attracted significant interest due to its close association with issues such as forest closeness centrality of networks (Jin et al., 2019; van der Grinten et al., 2021), determinantal point processes in machine learning (Kulesza et al., 2012), and multi-agent and network-based problems (Rossi et al., 2017). A nearly linear time algorithm combining the Johnson-Lindenstrauss lemma (Johnson & Lindenstrauss, 1984; Achlioptas, 2003) with a fast Laplacian solver was proposed in (Jin et al., 2019). This was followed by an approach in (van der Grinten et al., 2021) that integrated a single instance of the Laplacian solver with uniform spanning tree sampling. More recently, forest sampling algorithms introducing novel variance reduction techniques were developed, offering better theoretical guarantees than prior methods (Sun & Zhang, 2024).

However, when applied to signed graphs, where the forest matrix remains central to many problems (Halo et al., 2024; Li et al., 2022; Xu et al., 2020; Zhou et al., 2024), existing algorithms falter. This limitation stems from the fact that fast Laplacian solvers are not adaptable to signed contexts, and the traditional forest matrix theorem does not hold, rendering all forest sampling-based algorithms ineffective. Consequently, the introduction of a forest matrix theorem tailored for signed graphs, along with the development of a novel sampling-based method for efficiently estimating the diagonal of the forest matrix in such graphs, constitutes the primary focus of this paper.

## 3. Preliminaries

We define a directed signed graph $\mathcal{G} = (V, E, w)$ with $n = |V|$ nodes, $m = |E|$ edges, where $V = \{v_1, v_2, \ldots, v_n\}$ is the set of nodes, $E = \{(v_i, v_j) \in V \times V\}$ is the set of directed edges, and $w : E \mapsto \{+1, -1\}$ is the edge weight function, with the weight of an edge $e = (i, j)$ denoted by $w_{ij}$. We call an edge $e = (i, j)$ a positive (or negative) edge if its weight $w_{ij}$ is $+1$ (or $-1$). The edge sign represents the relationship between node $i$ and node $j$, which can be cooperative or competitive. In what follows, $v_i$ and $i$ are used interchangeably to represent node $v_i$ if incurring no confusion. A path $P$ from node $v_1$ to node $v_j$ is an alternating sequence of nodes and edges $v_1, (v_1, v_2), v_2, \ldots, v_{j-1}, (v_{j-1}, v_j), v_j$, where nodes are distinct. A loop is a path plus an arc from the ending node to the starting node.

For a directed signed graph $\mathcal{G} = (V, E, w)$, let $C = \{1, 2, \ldots, k\}$ be the cycle with nodes $1$ to $k$ and edges $(1, 2), (2, 3), \ldots, (k, 1)$. A cycle with only one node is called a trivial cycle. A non-trivial cycle is called nega-

tive (or positive) if the sign of the product of its arcs is negative (or positive). The graph $\mathcal{G} = (V, E, w)$ is defined as a balanced signed graph if either all its edges are positive or the vertices can be partitioned into two subsets such that each positive edge joins vertices in the same subset and each negative edge joins vertices in different subsets. Notably, balanced signed graphs do not contain any negative cycles.

Let $N(i)$ denote the set of nodes that can be accessed by node $i$. In other words, $N(i) = \{j : (i, j) \in E\}$. We define the degree of a node $i$ as $d_i = \sum_{j \in N(i)} |w_{ij}|$. We use a diagonal matrix $\boldsymbol{D} = \text{diag}\{d_1, d_2, \ldots, d_n\}$ to denote the degree matrix, and matrix $\boldsymbol{A} \in \mathcal{R}^{n \times n}$ to denote the signed adjacency matrix corresponding to the signed graph $\mathcal{G} = (V, E, w)$ with $\boldsymbol{A}_{ij} = w_{ij}$ for any edge $(i, j) \in E$, and $\boldsymbol{A}_{ij} = 0$ otherwise. Let $\boldsymbol{A}^{+} \in \mathcal{R}^{n \times n}$ be the positive adjacency matrix defined as $\boldsymbol{A}_{ij}^{+} = w_{ij}$ if $w_{ij} > 0$, and $\boldsymbol{A}_{ij}^{+} = 0$ otherwise. The negative adjacency matrix $\boldsymbol{A}^{-}$ is defined as $\boldsymbol{A}^{-} = \boldsymbol{A} - \boldsymbol{A}^{+}$. Then we define the signed Laplacian matrix as $\boldsymbol{L} = \boldsymbol{D} - \boldsymbol{A}$.

## 4. Forest Matrix Theorem on Signed Graphs

### 4.1. Forest Matrix on Signed Graphs

The forest matrix $\boldsymbol{Q} = (q_{ij})_{n \times n}$ is defined as $\boldsymbol{Q} = (\boldsymbol{I} + \boldsymbol{L})^{-1}$. The properties of the forest matrix in unsigned graphs have been extensively studied in (Chebotarev & Shamis, 1997; 1998; 2006; Sun & Zhang, 2023; 2024). For example, in unsigned directed graphs, the forest matrix is row stochastic, with all its components in the interval $[0, 1]$, and the diagonal elements in each column exceed the other elements.

In signed graphs, the forest matrix serves as the fundamental matrix in the signed opinion propagation Friedkin-Johnsen model (Xu et al., 2020; Halo et al., 2024). However, its properties differ from those in the unsigned case. The forest matrix is no longer row stochastic, and the non-diagonal elements may be less than zero. As we will show later, for any $i, j \in V$ with $i \neq j$, we have $0 \leq |q_{ij}| \leq q_{jj} \leq 1$.

### 4.2. Generalized Spanning Converging Forests

In this subsection, we introduce the concept of generalized spanning converging forests. A spanning subgraph of $\mathcal{G}$ is a subgraph of $\mathcal{G}$ with the node set being $V$ and the edge set being a subset of $E$. A generalized spanning converging forest is a spanning subgraph of $\mathcal{G}$, where the out-degree of each node is no more than 1, and all cycles are negative. Let $\mathcal{F}$ be the set of all generalized spanning converging forests of digraph $\mathcal{G}$. For any generalized spanning forest $\phi \in \mathcal{F}$, the root nodes of $\phi$ are those with an out-degree of 0. The root set $\mathcal{R}(\phi)$ is defined as $\mathcal{R}(\phi) = \{i : (i, j) \notin \phi,$ for any $j \in V\}$. We use $n^{-}(\phi)$ to denote the number of non-trivial

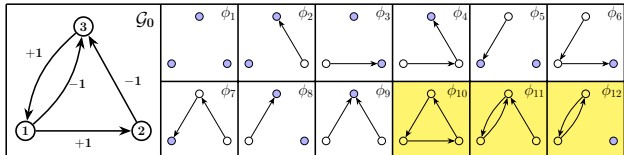

*Figure 1.* A toy signed graph $\mathcal{G}_0$ with its 12 generalized spanning converging forests. Blue nodes are roots.

negative cycles in $\phi$.

A generalized spanning converging forest $\phi$ may comprise several connected components. Let $\kappa(\phi)$ denote the number of connected components in $\phi$. By definition, each connected component in $\phi$ is either a rooted converging tree or a structure containing a negative cycle. Consequently, the relationship between the number of root nodes and the number of components is given by $|\mathcal{R}(\phi)| \leq \kappa(\phi) \leq n$. The lower bound, $|\mathcal{R}(\phi)| \leq \kappa(\phi)$, is achieved when there are no cycles within $\phi$. The upper bound, $\kappa(\phi) \leq n$, is reached when each node in $\phi$ is isolated, resulting in the absence of any edges within $\phi$. To effectively distinguish between the two possible scenarios within each connected component and to simplify notation, we define the function $r_\phi$ for each node $i$ in $\phi$ as follows: $r_\phi(i) = i$ if $i \in \mathcal{R}(\phi)$; $r_\phi(i) = 0$ if $i$ belongs to a cycle; otherwise, if $(i, j) \in E_\phi$ and $i$ does not belong to a cycle, we recursively define $r_\phi(i) = r_\phi(j)$.

From this definition, we observe that for any node $i \in \phi$, if the connected component containing $i$ includes a negative cycle, then $r_\phi(i) = 0$. Conversely, if the connected component containing $i$ is a rooted converging tree, then the function $r_\phi$ maps the node $i$ to its root in its connected component.

For nodes $i, j \in V$, define $\mathcal{F}_{ij}$ to be the set of those generalized spanning converging forests, where node $j$ is the root, and there is a path from node $i$ to node $j$. Then the function $r_\phi$ maps node $i$ to $j$, that is, $\mathcal{F}_{ij} = \{\phi : r_\phi(i) = j, \phi \in \mathcal{F}\}$. Then, for node $i \in V$, we have $\mathcal{F}_{ii} = \{\phi : i \in \mathcal{R}(\phi), \phi \in \mathcal{F}\}$. For a generalized spanning converging forest $\phi$, its weight $w(\phi)$ is defined as $w(\phi) = 2^{n^-(\phi)} \prod_{(i,j) \in E_\phi} |w_{ij}| = 2^{n^-(\phi)}$.

If there is no edge in $\phi$, its weight is defined to be 1. Define the weight of set $\mathcal{F}$ as $w(\mathcal{F}) = \sum_{\phi \in \mathcal{F}} w(\phi)$. Similarly, define $w(\mathcal{F}_{ii}) = \sum_{\phi \in \mathcal{F}_{ii}} w(\phi)$. There is something different when we define $w(\mathcal{F}_{ij}) = \sum_{\phi \in \mathcal{F}_{ij}} \text{sign}(P_{ij}) w(\phi)$, where $\text{sign}(P_{ij})$ is the sign of the product of the weights of the arcs in the path from node $i$ to node $j$ in $\phi$.

For example, we present a toy graph, $\mathcal{G}_0$, comprising 3 nodes and 4 edges, including two positive and two negative edges. We list all its 12 generalized spanning converging forests $\phi_1, \phi_2, \cdots, \phi_{12}$. Notably, the last three forests, highlighted with a yellow background in Figure 1, contain negative

cycles. Following the definition provided, the weight $w(\phi_i)$ is assigned as 1 for $i = 1, \cdots, 9$ and 2 for $i = 10, 11, 12$.

### 4.3. Signed Forest Matrix Theorem

In this subsection, we introduce the forest matrix theorem in signed graphs. We extend the forest matrix theorem from the unsigned case (Chebotarev & Shamis, 2006; 1997; 1998) to accommodate signed graphs. First, we propose two lemmas that establish the relationship between the determinant of the matrix $I + L$ and its submatrices, obtained by deleting one column and one row, with the weights of specific generalized spanning converging forests.

**Lemma 4.1.** *For a directed signed graph $\mathcal{G} = (V, E, w)$, the determinant of matrix $I + L$ is equal to the sum of the weights of all the generalized spanning converging forests:* $\det(I + L) = w(\mathcal{F})$.

**Lemma 4.2.** *For a directed signed graph $\mathcal{G} = (V, E, w)$, let $(I + L)_{-j,-i}$ denote the matrix obtained by deleting the $j$-th row and $i$-th column. Then the determinant of matrix $(I + L)_{-j,-i}$ is related to the generalized spanning converging forests as* $\det(I + L)_{-j,-i} = (-1)^{i+j} w(\mathcal{F}_{ij})$.

An illustrative example can be seen in Figure 1, where the determinant of the toy graph $\mathcal{G}_0$'s matrix $I + L$ is calculated to be 15, aligning with the combined weights of all generalized spanning converging forests: $\phi_1$ to $\phi_9$ each have a weight of 1, while $\phi_{10}$ to $\phi_{12}$ each have a weight of 2.

Building on these foundations, we can now state the Signed Forest Matrix Theorem:

**Theorem 4.3** (Signed Forest Matrix Theorem)**.** *For a directed signed graph $\mathcal{G} = (V, E, w)$, the entry of the forest matrix $Q = (I + L)^{-1} = (q_{ij})_{n \times n}$ is related to the generalized spanning converging forests as $q_{ij} = \frac{w(\mathcal{F}_{ij})}{w(\mathcal{F})}$.*

Theorem 4.3 shows that in a signed graph, the entry $q_{ij}$ of the forest matrix $Q$ represents the ratio of the sum of weights of the generalized spanning converging forests—where the root of node $i$ is node $j$ —relative to the sum of weights of all generalized spanning converging forests. Notably, when all edges are positive, this finding is consistent with the forest matrix theorem for unsigned graphs in prior studies (Chebotarev & Shamis, 2006; 1998). Building on the insights provided by Theorem 4.3, we introduce the following lemma, which details specific properties of the entries of the forest matrix in signed graphs:

**Lemma 4.4.** *For a signed graph $\mathcal{G} = (V, E, w)$, and any distinct nodes $i, j \in V$, the inequality $0 \leq |q_{ij}| \leq q_{jj} \leq 1$ holds. When $\mathcal{G}$ is a balanced signed graph, the sum of the absolute values of the entries in any row $i$ equals 1, that is $\sum_{j=1}^{n} |q_{ij}| = 1$. Moreover, in this scenario, the $i$-th diagonal element $q_{ii}$ satisfies $\frac{1}{1+d_i} \leq q_{ii} \leq \frac{2}{2+d_i}$.*

## 5. Positive Loop-Erased Random Walks

### 5.1. Generating a Generalized Spanning Converging Forest Based on Random Walk

In this subsection, we introduce a random walk approach to generate a generalized spanning converging forest on signed graphs. Before that, we briefly review the loop-erasure operation on a random walk (Lawler & Gregory, 1980), since it plays an important role in our algorithm. Concretely, for a random walk $P = v_1, (v_1, v_2), v_2, \ldots, v_{j-1}, (v_{j-1}, v_j), v_j$, the loop-erasure operation $P_{\text{LE}}$ on $P$ is an alternating sequence $\widetilde{v}_1, (\widetilde{v}_1, \widetilde{v}_2), \widetilde{v}_2 \ldots, \widetilde{v}_{q-1}, (\widetilde{v}_{q-1}, \widetilde{v}_q), \widetilde{v}_q$ of nodes and edges, which is obtained inductively as follows. First, set $\widetilde{v}_1 = v_1$ and append $\widetilde{v}_1$ to $P_{\text{LE}}$. Suppose that sequence $\widetilde{v}_1, (\widetilde{v}_1, \widetilde{v}_2), \widetilde{v}_2, \ldots, \widetilde{v}_{h-1}, (\widetilde{v}_{h-1}, \widetilde{v}_h), \widetilde{v}_h$ has been added to $P_{\text{LE}}$ for some $h \geq 1$. If $\widetilde{v}_h = v_j$, then $q = h$ and $\widetilde{v}_h$ is the last node in $P_{\text{LE}}$. Otherwise, define $\widetilde{v}_{h+1} = v_{r+1}$, where $r = \max\{i : v_i = \widetilde{v}_h\}$.

Wilson proposed an algorithm for generating a spanning tree rooted at a given node based on the loop-erasure operation on a random walk (Wilson, 1996). However, adapting the conventional loop-erased random walk method to generate a generalized spanning converging forest for signed graphs is challenging, owing to the differences between the signed forest matrix Theorem 4.3 and the unsigned case. Negative cycles are allowed in generalized spanning forests for signed graphs, and thus, the traditional loop-erased random walk approach requires modification. To address these challenges, we propose an extension of the traditional loop-erased random walk algorithm to generate a generalized spanning converging forest for signed graphs. Specifically, we describe the steps for generating a generalized spanning converging forest $\phi = (V_\phi, E_\phi)$ in a signed digraph $\mathcal{G} = (V, E, w)$ as follows:

(i) Set $\phi = (V_\phi, E_\phi) = (\emptyset, \emptyset)$.

(ii) Choose a node $i$ from $V \setminus V_\phi$ and create a random walk $P = v_i$ starting at node $i$ in $\mathcal{G}$.

(iii) At each time step, let $u$ denote the current node of the random walk $P$. The walk either terminates with probability $\frac{1}{1+d_u}$, in which case node $u$ is added to the set of root nodes of $\phi$, or jumps to a random neighbor $j$ of the current position of $u$. If the former case occurs, proceed to step (v). Otherwise, if the walk jumps from $u$ to $j$, add edge $(u, j)$ and node $j$ to $P$ and proceed to step (iv).

(iv) Suppose that now the random walk $P$ starts at node $i$ and ends at node $j$. If $j$ is already in the set $V_\phi$, proceed to step (v). Otherwise, check if there exists a negative cycle $C$ in $P$ that includes node $j$. If such a cycle is found, proceed to step (vi). Otherwise, continue the random walk according to step (iii).

(v) Perform loop-erasure operation on the random walk $P$

to get $P_{\text{LE}}$, and add the nodes and edges in $P_{\text{LE}}$ to $\phi$. Then update $V_\phi$ and $E_\phi$. If $V_\phi \neq V$, repeat step (ii); otherwise terminate the loop.

(vi) Assume that the current random walk $P$ goes from node $i$ to node $j$, and $j$ belongs to a negative cycle $C$. We can partition $P$ into two parts, namely $P'$ and $C$, where $P'$ is the portion of the walk preceding the negative cycle. The loop-erasure operation is then performed on the path $P'$ to obtain $P'_{\text{LE}}$, and the resulting path $(P'_{\text{LE}}, C)$, which connects the end of $P'_{\text{LE}}$ to the cycle $C$, is added to the graph $\phi$. The sets of vertices and edges in $\phi$, $V_\phi$ and $E_\phi$, are then updated accordingly. If $V_\phi \neq V$, the circulation starts again from step (ii); otherwise the algorithm terminates.

In Algorithm 1, we provide a detailed description of the pseudocode for algorithm GSCF. It is evident that this algorithm produces a generalized spanning converging forest. In the following subsection, we will delve into the algorithm's workings and prove that its expected running time is independent of the order in which nodes are selected.

## 5.2. Running Time Analysis

In this subsection, we present an analysis of the expected time complexity of Algorithm 1.

In Algorithm 1, each time a branch is added to $\phi$ in line 25 or 28, the random walk restarts from a new node by going back to line 4. Therefore, it is necessary to specify a predetermined order in which the nodes are selected in line 4 of the algorithm. In the following, we present a lemma, demonstrating that the expected time complexity of Algorithm 1 is independent of the order in which the nodes are selected in line 4 of the algorithm.

**Lemma 5.1.** *For a given graph $\mathcal{G} = (V, E, w)$, the expected time complexity of Algorithm 1 is independent of the order in which the random walk starts at each node.*

Next, we present Theorem 5.2, which provides insight into the expected time complexity of Algorithm 1.

**Theorem 5.2.** *The expected time complexity of Algorithm 1 is $O(n)$.*

## 6. Forest Sampling Algorithm for Estimating the Forest Matrix

### 6.1. Estimator for the Entry of Forest Matrix

In this subsection, we propose estimators for the entries of the forest matrix, leveraging the Signed Forest Matrix Theorem 4.3 and the positive loop-erased random walk introduced in Algorithm 1.

Consider a directed signed graph $\mathcal{G} = (V, E, w)$ with its corresponding forest matrix $Q$. Our goal is to give an es-

timation for $\widehat{q}_{ij}$ for the entry $q_{ij}$ for $i, j \in V$. Directly inverting the matrix $I + L$ to obtain $Q$ incurs a time complexity of $O(n^3)$, which is infeasible for large-scale graphs. According to Theorem 4.3, for any pair of nodes $i, j \in V$, the $(i, j)$-th element $q_{ij}$ of the forest matrix $Q$ can be represented as:

$$q_{ij} = \frac{w(\mathcal{F}_{ij})}{w(\mathcal{F})} = \frac{\sum_{\phi \in \mathcal{F}_{ij}} \text{sign}(P_{ij}) 2^{n^-(\phi)}}{\sum_{\phi \in \mathcal{F}} 2^{n^-(\phi)}}. \quad (1)$$

From Equation (1), the entries of the forest matrix in signed graphs can be interpreted as the ratio of the total weight of forests in $\mathcal{F}_{ij}$ to the total weight of all forests in $\mathcal{F}$. Specifically, $\mathcal{F}_{ij}$ consists of forests where nodes $i$ and $j$ belong to the same connected component, with $j$ serving as the root, that is $\mathcal{F}_{ij} = \{\phi : r_\phi(i) = j, \phi \in \mathcal{F}\}$. In Algorithm 1, a generalized spanning converging forest is generated using a positive loop-erased random walk. To estimate $q_{ij}$, we propose sampling $l$ forests using Algorithm 1. Before defining the estimator, we establish a lemma to demonstrate that the generalized spanning converging forests generated by Algorithm 1 are uniformly sampled from the set $\mathcal{F}$.

**Lemma 6.1.** *Suppose that $\phi_0 \in \mathcal{F}$ is a fixed generalized spanning converging forest, and Algorithm 1 returns a generalized spanning converging forest $\phi$. Then we have $\mathbb{P}(\phi = \phi_0) = \frac{1}{|\mathcal{F}|}$.*

With Lemma 6.1, now we suppose that we execute Algorithm 1 $l$ times to generate $l$ generalized spanning converging forests $\phi_1, \cdots, \phi_l$. Define the estimator $\widehat{w}_l(\mathcal{F})$ as $\widehat{w}_l(\mathcal{F}) = \frac{|\mathcal{F}|}{l} \sum_{k=1}^l 2^{n^-(\phi_k)}$. And define the estimator $\widehat{w}_l(\mathcal{F}_{ij})$ as $\widehat{w}_l(\mathcal{F}_{ij}) = \frac{|\mathcal{F}|}{l} \sum_{k=1}^l 2^{n^-(\phi_k)} \text{sign}(P_{ij}) \mathbb{I}_{\{r_{\phi_k}(i)=j\}}$. Here, $\mathbb{I}$ denotes the indicator function, and $\mathbb{I}_{\{r_{\phi_k}(i)=j\}}$ takes the value 1 if node $j$ is the root in forest $\phi_k$, and 0 otherwise. Then we have the following lemma:

**Lemma 6.2.** *For nodes $i, j \in V$ and $l$ generalized spanning converging forests generated from Algorithm 1, the variables $\widehat{w}_l(\mathcal{F})$ and $\widehat{w}_l(\mathcal{F}_{ij})$ are unbiased estimators of $w(\mathcal{F})$ and $w(\mathcal{F}_{ij})$, respectively.*

According to Lemma 6.2 and equation (1), we can rewrite the entry $q_{ij}$ as $q_{ij} = \mathbb{E}(\widehat{w}_l(\mathcal{F}_{ij})) / \mathbb{E}(\widehat{w}_l(\mathcal{F}))$. For each pair of nodes $i, j \in V$, we define the variable $\widehat{q}_{ij} = \widehat{w}_l(\mathcal{F}_{ij}) / \widehat{w}_l(\mathcal{F})$. Using this sampling-based estimator, we can efficiently approximate the entries of the forest matrix $Q$, as well as compute linear combinations of its entries.

### 6.2. Estimation for the Diagonal of Forest Matrix

In this subsection, we propose an efficient algorithm for estimating the diagonal vector of the forest matrix, which is denoted as $q = (q_{11}, \cdots, q_{nn})^\top$.

The diagonal elements of the forest matrix are significant as node centrality measures in unsigned graphs (Jin et al., 2019; Sun & Zhang, 2023). In the context of signed graphs, the diagonal entry $q_{ii}$ remains important. Specifically, $q_{ii}$ represents the ratio of the total weight of generalized spanning converging forests rooted at node $i$ to the total weight of all forests in the graph, as shown in Theorem 4.3. Unlike in unsigned graphs, where the lower bound of $q_{ii}$ is $\frac{1}{1+d_i}$, in signed graphs, $q_{ii}$ can be smaller but remains strictly positive. Furthermore, while off-diagonal entries $q_{ij}$ of the forest matrix can take both positive and negative values, the diagonal entries $q_{ii}$ are always positive, as established in Lemma 4.4.

Signed graphs exhibit a more complex structure due to the presence of negative edges. As a result, existing methods for unsigned graphs (Jin et al., 2019; van der Grinten et al., 2021; Sun & Zhang, 2024) fail to extend effectively to signed graphs. To address this limitation, we use the vector $\widehat{\boldsymbol{q}} = (\widehat{q}_{11}, \cdots, \widehat{q}_{nn})^\top$, to estimate the diagonal vector $\boldsymbol{q}$ of the forest matrix, where each diagonal entry is estimated as $\widehat{q}_{ii} = \widehat{w}_l(\mathcal{F}_{ii})/\widehat{w}_l(\mathcal{F})$. To efficiently estimate the diagonal vector $\boldsymbol{q}$ of the forest matrix $\boldsymbol{Q}$, we sample $l$ generalized spanning converging forests to compute the estimator $\widehat{\boldsymbol{q}}$.

However, because the estimator $\widehat{\boldsymbol{q}}$ primarily focuses on identifying root nodes, it may overlook additional informative aspects of the network structure. We now propose an alternative expression for the diagonal element $q_{ii}$. Leveraging the equation $\boldsymbol{Q}(\boldsymbol{I} + \boldsymbol{L}) = \boldsymbol{I}$, we obtain that for any node $i \in V$, $1 = (1 + d_i)q_{ii} - \sum_{k \neq i} q_{ik}w_{ki}$. That is, $q_{ii} = \frac{1}{1+d_i}(1 + \sum_{k \neq i} q_{ik}w_{ki})$. Accordingly, we define $\widetilde{q}_{ii}$ as $\widetilde{q}_{ii} = \frac{1}{1+d_i}(1 + \sum_{k \neq i} \widehat{q}_{ik}w_{ki})$, and we use $\widetilde{q}_{ii}$ to estimate $q_{ii}$. Below, we outline the pseudocode for our two algorithms, named FOREST MATRIX DIAGONAL ESTIMATOR (FMDE) and FOREST MATRIX DIAGONAL ESTIMATOR PLUS (FMDE+). The pseudocodes are provided in Appendix.

We now introduce Theorem 6.3 that highlights the efficiency of the estimator $\widetilde{q}_{ii}$ compared to $\widehat{q}_{ii}$ within the context of balanced signed graphs.

**Theorem 6.3.** *In a balanced signed graph $\mathcal{G}$, the variance of estimator $\widetilde{q}_{ii}$ is lower than that of $\widehat{q}_{ii}$. This suggests that, for a fixed number of samples $l$, $\widetilde{q}_{ii}$ is likely to yield a closer approximation to the true value $q_{ii}$ than $\widehat{q}_{ii}$.*

Theorem 6.3 theoretically demonstrates that in balanced signed graphs, the FMDE+ algorithm outperforms FMDE due to its use of estimators with reduced variance. In the experimental section below, we will show that FMDE+ also achieves superior accuracy compared to FMDE even in unbalanced signed graphs.

Utilizing Theorem 5.2, the time complexity of Algorithm 2 is $O(ln)$, where $l$ is the number of generalized spanning

converging forests. As we increase the number of sampled forests $l$, we observe a corresponding decrease in the estimation error between $\widehat{\boldsymbol{q}}[i]$ and the actual value $q_{ii}$. To quantify this relationship, we introduce Theorem 6.4, which specifies the necessary size of $l$ to achieve a necessary error guarantee with a high probability.

**Theorem 6.4.** *Define $\alpha = \max\{2^{n^-(\phi)} : \phi \in \mathcal{F}\}$ and $\beta = \sum_{\phi \in \mathcal{F}} 2^{n^-(\phi)}/|\mathcal{F}|$. For any node $i \in V$, and parameters $\epsilon, \delta \in (0, 1)$, if $l$ is chosen obeying $l = \left\lceil \frac{1}{2} \frac{\alpha^2}{\beta^2} (\frac{\epsilon+2}{\epsilon})^2 \log(\frac{2}{\delta}) \right\rceil$, then the following inequalities hold with probability at least $1 - \delta$:*

$$\mathbb{P}(|\widehat{w}_l(\mathcal{F}) - w(\mathcal{F})| \geq |\mathcal{F}|\frac{\epsilon\beta}{2+\epsilon}) < \delta. \qquad (2)$$

$$\mathbb{P}(|\widehat{w}_l(\mathcal{F}_{ii}) - w(\mathcal{F}_{ii})| \geq |\mathcal{F}|\frac{\epsilon\beta}{2+\epsilon}) < \delta. \qquad (3)$$

*If the following inequalities hold, then the approximation $\widehat{\boldsymbol{q}}[i]$ of $q_{ii}$ returned by Algorithm 2 satisfies the following relation: $q_{ii} - \epsilon \leq \widehat{\boldsymbol{q}}[i] \leq q_{ii} + \epsilon$.*

In real-life networks, the percentage of negative edges is extremely small (Chiang et al., 2014). Moreover, it is believed that a signed social network evolves towards a balanced state; otherwise, a state of unbalance will produce tension (Singh & Adhikari, 2017). Note that when there are no negative edges or $\mathcal{G}$ constitutes a balanced signed graph, the factor $\alpha/\beta$ equals 1. Consequently, according to Theorem 6.4, the required sample size $l$ will not become excessively large due to an expansion in the ratio $\alpha/\beta$. This ensures that our sampling algorithm remains efficient and practical for applications in real-life networks.

### 6.3. Expressed Opinion Estimation in Signed Friedkin-Johnsen Model

The Friedkin-Johnsen (FJ) model is a popular model for analyzing opinion evolution and formation on graphs (Friedkin & Johnsen, 1990; Bindel et al., 2015; He et al., 2020; Rahaman & Hosein, 2021). In the signed FJ model, each node $i \in V$ is associated with two types of opinions: the internal opinion and the expressed opinion. In the signed FJ model, each node $i \in V$ has an internal opinion $s_i \in [-1, 1]$ and an expressed opinion $z_i(t)$ at time $t$. At time $t + 1$, the expressed opinion evolves according to $z_i(t + 1) = \frac{1}{1+d_i}(s_i + \sum_{j \in N(i)} w_{ij}z_j(t))$. Let $\boldsymbol{s} = (s_1, s_2, \ldots, s_n)^\top$ be the internal opinion vector. The expressed opinion vector converges to an equilibrium $\boldsymbol{z} = (z_1, z_2, \ldots, z_n)^\top$ satisfying $\boldsymbol{z} = (\boldsymbol{I} + \boldsymbol{L})^{-1}\boldsymbol{s} = \boldsymbol{Q}\boldsymbol{s}$.

While the signed FJ model has been widely studied (Xu et al., 2020; Rahaman & Hosein, 2021; He et al., 2020; 2022; Tang et al., 2016; Halo et al., 2024), efficient algorithms for estimating expressed opinions in directed signed graphs are lacking due to the difficulty of estimating the forest matrix.

Using the predefined estimator, we can approximate the $i$-th expressed opinion $z_i$. Since $z_i = \sum_{j=1}^{n} q_{ij} s_j$ and $\widehat{q}_{ij}$ serves as an estimator for $q_{ij}$, we define $\widehat{z}_i = \sum_{j=1}^{n} \widehat{q}_{ij} s_j$ as the estimator for $z_i$. Specifically, we first sample a set of $l$ generalized spanning converging forests, stored in a forest list $L$, using Algorithm 1. This sampling procedure incurs a time and space complexity of $O(ln)$.

To estimate the expressed opinion of a specific node, we traverse the forest list to compute $\widehat{z}_i$, which requires only $O(l)$ time. This process is detailed in the following algorithm, FJOE (Friedkin-Johnsen Opinion Estimation). Notably, while the internal opinion vector $s$ may change, resampling the forest is unnecessary as long as the graph structure remains unchanged. This allows our algorithm to efficiently query the expressed opinion.

By setting $l = O(\frac{\alpha^2}{\beta^2} \cdot \frac{1}{\epsilon^2} \log\left(\frac{1}{\delta}\right))$, following the approach in Theorem 6.4, we can guarantee that the estimation error satisfies $|\widehat{z}_i - z_i| \le \epsilon$ with a probability of at least $1 - \delta$.

# 7. Experiments

## 7.1. Setup

**Dataset.** The datasets of selected real networks are publicly available in the KONECT (Kunegis, 2013) and SNAP (Leskovec & Sosič, 2016). Our experiments are conducted on a diverse range of networks. Details of these datasets are presented in Table 3. We utilize both original signed graphs and modified signed graphs for our experiments. The modified signed graphs are generated from real unsigned graphs by randomly assigning a negative sign to each edge with a probability of 0.2. These modified signed graphs are denoted with a superscript asterisk in Table 3.

**Algorithms.** To evaluate the performance of our algorithms in estimating the diagonal elements for forest matrix of signed graphs, we compare our two proposed algorithms, FMDE and FMDE+, against the ground truth, which is obtained by directly inverting the matrix $I + L$. Additionally, we evaluate the accuracy of FJOE by performing 100 random queries and comparing the results with the ground truth.

## 7.2. Forest Matrix Diagonal Estimation

### 7.2.1. ACCURACY

We first evaluate the accuracy of our algorithms FMDE and FMDE+ with the ground truth. To this end, we conduct experiments on six small-sized networks, as obtaining the ground truth by inverting the matrix $I + L$ is computationally intensive and memory-consuming for larger graphs. The details of these networks: Adolescent*, Bitcoinotc, Gnutella08*, Wikielec, Wikipedia*, and SlashdotZoo are

listed in Table 3. Of these, three are original signed graphs, while the remaining three, marked with a superscript asterisk, are modified signed graphs.

To evaluate the accuracy of our two algorithms, we use the average relative error across all nodes. For each signed graph $\mathcal{G} = (V, E, w)$, we initially compute the forest matrix $Q = (I + L)^{-1}$ to obtain its diagonal $q$. Our algorithms, FMDE and FMDE+, then estimate the diagonal, resulting in $\widehat{q}$ and $\widetilde{q}$, respectively. The average relative error for algorithm FMDE is calculated using $\frac{1}{n} \sum_{i=1}^{n} \frac{|q_i - \widehat{q}_i|}{q_i}$, and similarly for FMDE+. We set $\epsilon = 0.1, 0.2, 0.3$ to examine performance under these settings, with the results depicted in Figure 2.

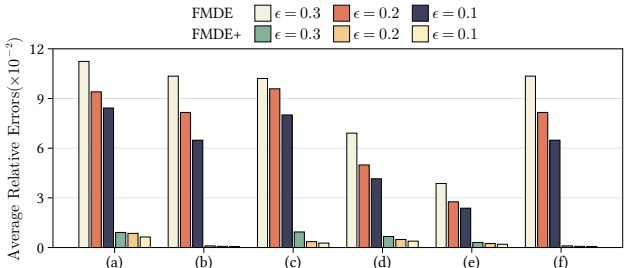

*Figure 2.* Comparison of average relative errors of the diagonals for algorithms FMDE and FMDE+ on six graphs: Bitcoinotc(a), Wikielec(b), SlashdotZoo(c), Adolescent*(d), Gnutella08*(e), Wikipedia*(f) across three different settings of $\epsilon$.

The results displayed in Figure 2 demonstrate that as $\epsilon$ decreases, the number of samples increases, which consequently reduces the average relative error. This trend is consistent for both algorithms, FMDE and FMDE+. Furthermore, despite the distinction between original signed graphs (a), (b), (c) and modified signed graphs (d), (e), (f), the performance outcomes are comparably robust. Notably, FMDE+ significantly outperforms FMDE in terms of accuracy, achieving results approximately ten times better. Specifically, the average relative error for FMDE+ remains below 0.01 across all tested graphs. In particular instances, such as in graphs (b) and (f), the error margin even drops below 0.001 for all three $\epsilon$ settings. This marked improvement is attributed to the enhancements incorporated in FMDE+, which employs a superior estimator as theoretically detailed in previous sections. In conclusion, the results returned by the FMDE+ algorithm are more convincing and exhibit high accuracy.

### 7.2.2. EFFICIENCY AND SCALABILITY

We now demonstrate that our algorithms, FMDE and FMDE+, are more efficient than the direct matrix inversion method, referred to here as EXACT. To illustrate this, Table 1 compares the performance of EXACT, FMDE, and FMDE+. The results indicate that for the first six small-sized graphs, both FMDE and FMDE+ significantly

outperform EXACT in terms of computational speed for all three $\epsilon$ settings chosen. Furthermore, it is observed that as $\epsilon$ decreases, the running time increases. Besides, algorithm FMDE+, requires slightly more time than FMDE for a fixed $\epsilon$ due to its need to collect additional information, as outlined in Algorithm 2.

*Table 1.* Running time (seconds) of algorithms EXACT, FMDE and FMDE+.

| Network | | Time(seconds) | | | | | |
|---|---|---|---|---|---|---|---|
| | EXACT | FMDE | | | FMDE+ | | |
| | | $\epsilon = 0.3$ | 0.2 | 0.1 | 0.3 | 0.2 | 0.1 |
| Adolescent | 0.30 | 0.019 | 0.023 | 0.031 | 0.046 | 0.063 | 0.095 |
| Bitcoinotc | 1.87 | 0.039 | 0.055 | 0.070 | 0.072 | 0.118 | 0.165 |
| Gnutella08 | 2.28 | 0.025 | 0.041 | 0.060 | 0.032 | 0.044 | 0.064 |
| Wikielec | 3.01 | 0.031 | 0.045 | 0.062 | 0.041 | 0.052 | 0.098 |
| Wikipedia | 22.96 | 0.066 | 0.106 | 0.131 | 0.410 | 0.677 | 0.874 |
| SlashdotZoo | 991.5 | 0.361 | 0.464 | 0.791 | 0.487 | 0.833 | 1.033 |
| Epinions | - | 0.377 | 0.587 | 0.854 | 0.345 | 0.616 | 1.141 |
| WikiL | - | 0.701 | 1.535 | 2.036 | 0.732 | 1.577 | 2.222 |
| Youtube | - | 4.656 | 10.19 | 11.45 | 6.378 | 13.68 | 22.13 |
| Dblp | - | 53.55 | 168.8 | 269.3 | 108.2 | 325.5 | 518.0 |
| Livejournal | - | 459.0 | 1031 | 1623 | 604.1 | 1311 | 2114 |
| FullUSA | - | 918.8 | 1832 | 2803 | 1204 | 2015 | 3598 |

However, for the large graphs EXACT is unable to execute due to time and memory constraints. In contrast, FMDE and FMDE+ continue to perform efficiently on these networks. Notably, both algorithms are scalable to massive networks with more than twenty million nodes, such as FullUSA, which has over $2.3 \times 10^7$ nodes. Remarkably, both algorithms deliver results for our three $\epsilon$ settings within at most one hour. Thus, FMDE and FMDE+ not only provide accurate estimation of the diagonal elements of the forest matrix but also demonstrate remarkable efficiency and scalability to extensive graph sizes.

### 7.3. Opinion Estimation in Signed FJ Model

In this subsection, we evaluate the accuracy of FJOE by performing 100 random queries and comparing the results with the ground truth. We vary the $\epsilon$ values at 0.3, 0.2, and 0.1, and determine $l$ based on Theorem 6.4. The FJOE algorithm requires pre-sampling of $l$ generalized spanning converging forests, and its running time is comparable to that of FMDE, as shown in Table 1. The ground truth is obtained via matrix inversion, with computational time similar to that of EXACT in Table 1. We present the running time of FJOE along with the average absolute error of the opinions computed over 100 random queries. The details are summarized in Table 2.

Table 2 shows that as $\epsilon$ decreases, the corresponding $l$ increases, leading to smaller absolute errors. Our algorithm, FJOE, operates with a time complexity of $O(l)$ and demonstrates exceptionally fast performance. For instance, on the largest graph, FullUSA, the running time is less than

*Table 2.* Running time($\times 10^{-4}$ seconds) and absolute error ($\times 10^{-2}$) of algorithm FJOE

| Network | Time for FJOE | | | Absolute Error | | |
|---|---|---|---|---|---|---|
| | $\epsilon =0.3$ | 0.2 | 0.1 | 0.3 | 0.2 | 0.1 |
| Adolescent | 1.1 | 3.2 | 7.3 | 2.1 | 1.5 | 0.9 |
| Bitcoinotc | 1.2 | 4.2 | 9.5 | 3.2 | 2.3 | 1.6 |
| Gnutella08 | 1.1 | 3.2 | 8.5 | 0.7 | 0.2 | 0.1 |
| Wikielec | 2.0 | 5.2 | 7.4 | 1.2 | 0.9 | 0.7 |
| Wikipedia | 2.5 | 6.1 | 8.2 | 3.1 | 2.9 | 1.5 |
| SlashdotZoo | 2.1 | 5.0 | 7.2 | 2.1 | 1.2 | 0.4 |
| Epinions | 2.4 | 4.9 | 8.2 | - | - | - |
| WikiL | 1.9 | 5.1 | 9.5 | - | - | - |
| Youtube | 2.5 | 6.5 | 10.2 | - | - | - |
| Dblp | 2.8 | 5.8 | 9.8 | - | - | - |
| Livejournal | 2.7 | 6.0 | 10.2 | - | - | - |
| FullUSA | 2.9 | 7.2 | 11.5 | - | - | - |

$3 \times 10^{-4}$ seconds when $\epsilon = 0.3$. This indicates that the algorithm's efficiency remains largely unaffected by the growth in graph size. Furthermore, the absolute error remains under $2 \times 10^{-2}$ for the first six graphs when $\epsilon = 0.1$, demonstrating that the algorithm achieves high accuracy even on diverse network structures.

## 8. Conclusions

In this paper, we addressed the problem of fast estimation of the forest matrix of a signed graph. We first introduced the signed forest matrix theorem, which provides crucial insights into the properties of the forest matrix. Then we proposed a novel algorithm GSCF for generating generalized spanning converging forests, serving as the cornerstone for subsequent algorithms. Furthermore we developed two efficient sampling algorithms, FMDE and FMDE+, designed to estimate the diagonal of the forest matrix. FMDE+, in particular, incorporates more comprehensive information, resulting in superior performance both theoretically and experimentally. We also proposed an algorithm FJOE to estimate the expressed opinion of individuals in the signed FJ model. Finally, we conducted extensive experiments on various signed graphs, which demonstrated that our algorithms are not only effective and efficient but also scalable to massive networks with more than twenty million nodes.

In future work, we aim to extend our algorithms to address additional challenges on signed graphs, including optimization problems related to the forest matrix and graph embedding tasks for signed networks. These extensions will further enhance the utility of our framework for understanding and modeling complex signed interactions in large-scale social and information networks.

## Acknowledgements

The work was supported by the National Natural Science Foundation of China (Nos. 62372112 and 61872093).

## Impact Statement

This paper advances the efficient analysis of signed graphs and signed network models. The proposed methods provide scalable tools for estimating forest matrix quantities and studying opinion dynamics in networks with positive and negative relationships. We do not identify specific ethical or societal risks beyond those generally associated with machine learning, network analysis, and computational social science.

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

# A. Proofs

In this section, we provide proofs of selected lemmas and theorems.

## A.1. Proof of Lemma 4.1

**Proof.** We define the function $\pi : V \mapsto V$ as a permutation of the node set $V = \{1, \ldots, n\}$, and use $\mathcal{P}(V)$ to denote the set of all permutations of set $V$. We use $N(\pi)$ to denote the inversion number of $\pi$, that is $N(\pi) = |\{(i, j) : i < j, \pi(i) > \pi(j)\}|$. Each permutation $\pi$ can be decomposed into disjoint cycles $C_1, C_2 \ldots, C_{n(\pi)}$, where $n(\pi)$ represents the number of cycles in the decomposition. Let $n^-(\pi)$ and $n^+(\pi)$ denote the number of non-trivial negative and positive cycles in $\pi$, respectively.

From the definition of determinant, we obtain that

$$\det(\boldsymbol{I} + \boldsymbol{L}) = \sum_{\pi \in \mathcal{P}(V)} (-1)^{N(\pi)} \prod_{i \in V, \pi(i)=j} \boldsymbol{e}_i^\top (\boldsymbol{I} + \boldsymbol{L}) \boldsymbol{e}_j. \tag{4}$$

For a cycle $C_i$ belonging to $\pi$, its inversion number is $|C_i| - 1$. Then we have $(-1)^{N(\pi)} = \prod_{k=1}^{n(\pi)} (-1)^{|C_k|-1}$. Rewriting the determinant we obtain:

$$= \sum_{\pi \in \mathcal{P}(V)} \prod_{k=1}^{n(\pi)} (-1)^{|C_k|-1} \prod_{i:\pi(i)=i} \left(1 + \sum_{j \neq i} |w_{ij}|\right) \prod_{i:\pi(i)=j, i \neq j} (-w_{ij}). \tag{5}$$

We simplify the product terms further:

$$= \sum_{\pi \in \mathcal{P}(V)} (-1)^{n^-(\pi)+n^+(\pi)} \prod_{i:\pi(i)=i} \left(1 + \sum_{j \neq i} |w_{ij}|\right) \prod_{i:\pi(i)=j, i \neq j} w_{ij}$$

$$= \sum_{\pi \in \mathcal{P}(V)} (-1)^{n^+(\pi)} \prod_{i:\pi(i)=i} \left(1 + \sum_{j \neq i} |w_{ij}|\right) \prod_{i:\pi(i)=j, i \neq j} |w_{ij}| \tag{6}$$

For a permutation $\pi$, let $P(\pi) = \{i \in V : \pi(i) = i\}$ be the set of fixed points. We now define a set of mappings $\mathcal{M}(\pi)$. For a mapping $\widehat{\pi} \in \mathcal{M}(\pi), \widehat{\pi} : V \mapsto V$, it satisfies

$$\widehat{\pi}(i) = \begin{cases} j & i \in P(\pi), j \in \{i\} \cup N_i, \\ \pi(i) & i \notin P(\pi). \end{cases} \tag{7}$$

For each permutation $\pi \in \mathcal{P}(V)$ and corresponding mapping $\widehat{\pi} \in \mathcal{M}(\pi)$, we define an induced spanning subgraph $\widehat{\mathcal{G}}(\widehat{\pi}) = (V, E(\widehat{\pi}), w)$, where $E(\widehat{\pi}) = (i, j) : \widehat{\pi}(i) = j, i \neq j, i \in V$. We can then express the determinant as follows:

$$\det(\boldsymbol{I} + \boldsymbol{L}) = \sum_{\pi \in \mathcal{P}(V)} \sum_{\widehat{\pi} \in \mathcal{M}(\pi)} (-1)^{n^+(\pi)} \prod_{i:\widehat{\pi}(i)=j, i \neq j} |w_{ij}|$$

$$= \sum_{\pi \in \mathcal{P}(V)} \sum_{\widehat{\pi} \in \mathcal{M}(\pi)} (-1)^{n^+(\pi)}. \tag{8}$$

We then rearrange the sum order of $\pi$ and $\widehat{\pi}$:

$$= \sum_{\widehat{\pi}} \sum_{\pi:\widehat{\pi} \in \mathcal{M}(\pi)} (-1)^{n^+(\pi)}. \tag{9}$$

For any non-trivial cycle in $\widehat{\mathcal{G}}(\widehat{\pi})$, it either belongs to the decomposition of $\pi$ or not. Let $n^+(\widehat{\pi})$ and $n^-(\widehat{\pi})$ denote the number of non-trivial positive and negative cycles in $\widehat{\mathcal{G}}(\widehat{\pi})$, respectively. Summing over the non-trivial positive and negative cycles in the decompositions, we find

$$\sum_{\pi:\widehat{\pi} \in \mathcal{M}(\pi)} (-1)^{n^+(\pi)} = \sum_{i=0}^{n^+(\widehat{\pi})} \binom{n^+(\widehat{\pi})}{i} (-1)^i \sum_{j=0}^{n^-(\widehat{\pi})} \binom{n^-(\widehat{\pi})}{j}$$

$$= (1-1)^{n^+(\widehat{\pi})} (1+1)^{n^-(\widehat{\pi})} = \begin{cases} 0 & n^+(\widehat{\pi}) \neq 0, \\ 2^{n^-(\widehat{\pi})} & n^+(\widehat{\pi}) = 0. \end{cases} \tag{10}$$

This implies that for a fixed $\widehat{\pi}$, the expression $\sum_{\pi:\widehat{\pi}\in\mathcal{M}(\pi)}(-1)^{n^+(\pi)}$ equals $2^{n^-(\widehat{\pi})}$ if and only if $n^+(\widehat{\pi}) = 0$. In this scenario, the induced graph $\widehat{\mathcal{G}}(\widehat{\pi})$ corresponds to the generalized spanning converging forest previously defined. Hence, we conclude

$$\det(\boldsymbol{I} + \boldsymbol{L}) = \sum_{\widehat{\pi}:n^+(\widehat{\pi})=0} 2^{n^-(\widehat{\pi})} = \sum_{\phi \in \mathcal{F}} w(\phi) = w(\mathcal{F}), \tag{11}$$

which finishes the proof. $\square$

## A.2. Proof of Lemma 4.2

**Proof.** Similarly to the proof of Lemma 4.1, we now define the function $\pi$ as a bijection from the node set $V \setminus \{j\}$ to the node set $V \setminus \{i\}$. We use $N(\pi)$ to denote the inversion number of $\pi$. Notice that the permutation $\pi$ can be decomposed into a path $P_{ij}$ from node $i$ to node $j$ and disjoint cycles $C_1, C_2 \ldots, C_{n(\pi)}$, where $n(\pi)$ denotes the number of cycles in the decomposition. Let $n(P_{ij})$ be the number of nodes in $P_{ij}$. Let $n^-(\pi)$ and $n^+(\pi)$ be the number of non-trivial negative and positive cycles of $\pi$ respectively. We use $\text{sign}(P_{ij})$ to denote the sign of the product of the arcs in the path $P_{ij}$. Then we obtain that,

$$\prod_{i:\pi(i)=j,i\neq j} w_{ij} = \text{sign}(P_{ij})(-1)^{n^-(\pi)} \prod_{i:\pi(i)=j,i\neq j} |w_{ij}|. \tag{12}$$

To obtain the inversion number of $\pi$, we first define a mapping $\pi'$ mapping node $j$ to node $i$. Then the mapping $\pi \oplus \pi'$ is a permutation of set $V$. Thus one obtains that

$$(-1)^{N(\pi\oplus\pi')} = (-1)^{n(P_{ij})} \prod_{k=1}^{n(\pi)} (-1)^{|C_k|-1}. \tag{13}$$

Since the change of inversion number after adding $\pi'$ has the same parity as $i + j - 1$, one has

$$(-1)^{N(\pi)} = (-1)^{i+j}(-1)^{n(P_{ij})-1} \prod_{k=1}^{n(\pi)} (-1)^{|C_k|-1}. \tag{14}$$

Following similar steps in the proof of Lemma 4.1, one gets that

$$\det(\boldsymbol{I} + \boldsymbol{L})_{-j,-i} = (-1)^{i+j} \sum_{\phi \in \mathcal{F}_{ij}} \text{sign}(P_{ij})w(\phi) = (-1)^{i+j}w(\mathcal{F}_{ij}), \tag{15}$$

which completes the proof. $\square$

## A.3. Proof of Lemma 4.4

**Proof.** According to Theorem 4.3, it is straightforward to derive that for any distinct nodes $i, j \in V$, the inequality $0 \leq |q_{ij}| \leq q_{jj} \leq 1$ holds. In scenarios where $\mathcal{G} = (V, E, w)$ constitutes a balanced signed graph, the graph contains no non-trivial negative cycles. Under such circumstances, the path sign between any pair of nodes $i, j \in V$ is uniformly positive or negative, leading to the equation $\sum_{j=1}^{n} |q_{ij}| = \frac{\sum_{j=1}^{n} |w(\mathcal{F}_{ij})|}{w(\mathcal{F})} = 1$. Moreover, leveraging the relation $\boldsymbol{Q}(\boldsymbol{I} + \boldsymbol{L}) = \boldsymbol{I}$, we obtain that for any node $i \in V$, $1 = (1 + d_i)q_{ii} - \sum_{k\neq i} q_{ik}w_{ki}$. That is, $q_{ii} = \frac{1}{1+d_i}(1 + \sum_{k\neq i} q_{ik}w_{ki}) \leq \frac{1}{1+d_i}(1 + \sum_{k\neq i} |q_{ik}|) = \frac{1}{1+d_i}(1 + q_{ii})$, which can be simplified to $q_{ii} \leq \frac{2}{2+d_i}$. Moreover, in this case, $q_{ik}w_{ki}$ must be non-negative, leading to the fact that $q_{ii} \geq \frac{1}{1+d_i}$, which finishes the proof. $\square$

## A.4. Proof of Theorem 4.3

**Proof.** According to Lemma 4.1 and Lemma 4.2, we obtain that

$$q_{ij} = \frac{(-1)^{i+j}\det(\boldsymbol{I} + \boldsymbol{L})_{-j,-i}}{\det(\boldsymbol{I} + \boldsymbol{L})} = \frac{w(\mathcal{F}_{ij})}{w(\mathcal{F})}, \tag{16}$$

which finishes the proof. $\square$

## A.5. Proof of Lemma 5.1

**Proof.**

We first introduce some notations. For a signed graph $\mathcal{G} = (V, E, w)$ and a node $i \in V$, we define $t_i$ as a random variable that takes values from the set $\{-1\} \cup N(i)$, where the probability of $t_i = -1$ is $\frac{1}{1+d_i}$, and the probability of $t_i = u$ for any node $u \in N(i)$ is also $\frac{1}{1+d_i}$. Then we define a matrix $\boldsymbol{T}^L = (t_{ij}^L)_{n \times L}$. The entry $t_{ij}^L$ in row $i$ and column $j$ of the matrix $\boldsymbol{T}^L$ is a random variable that is independently and identically distributed with $t_i$.

We can utilize the matrix $\boldsymbol{T}^L$ to determine the next node to visit during the random walk process in Algorithm 1. To be more specific, we begin by defining a vector $\boldsymbol{h} = (h_i)_{n \times 1}$, where $h_i$ is initialized to 1 at the start of our algorithm. During the random walk process, suppose the walk is currently at node $i$, and we need to select the next target node. We set $j = h_i$, and then look at the $j$-th column of the matrix $\boldsymbol{T}^L$ corresponding to node $i$. The entry $t_{ij}^L$ in this column represents the next node to visit. If $t_{ij}^L = -1$, we designate node $i$ as the new root node. Otherwise, if $t_{ij}^L = u$, where $u$ is a node adjacent to $i$, we proceed to node $u$ for the next step of the walk. After selecting the next target node, we update $h_i$ to $h_i + 1$. When Algorithm 1 terminates, we obtain a vector $\boldsymbol{h}$. We can measure the time complexity of Algorithm 1 by computing the $\ell_1$-norm of $\boldsymbol{h}$, denoted by $\|\boldsymbol{h}\|_1$, which is simply the sum of all elements in $\boldsymbol{h}$, i.e., $\sum_{i=1}^{n} h_i$.

In Algorithm 1, we perform the loop-erasure operation if a non-trivial positive cycle exists. A cycle with the same nodes may be traversed several times during the algorithm so that it may be erased many times. However, since we use matrix $\boldsymbol{T}^L$ to determine the next node to visit, every entry in matrix $\boldsymbol{T}^L$ can only form one positive cycle and be erased once.

To denote the cycle $C$ and its position in matrix $\boldsymbol{T}^L$, we use the $n$-dimensional vector $\boldsymbol{c} = (c_1, \cdots, c_n)^\top$. For any $i \in V$, we have $c_i \in \{0, 1, \cdots, L\}$. If $c_i \neq 0$, it means that node $i$ is in the cycle and vice versa. To be more specific, $C$ is composed of edges $(i, t_{ic_i}^L)$ for any node $i$ that satisfies $c_i \neq 0$. That is, $C = \bigcup_{i:c_i \neq 0}(i, t_{ic_i}^L)$.

Consider two different permutations of the node set $V$, denoted as $\pi_1$ and $\pi_2$. Given a fixed matrix $\boldsymbol{T}^L$ with sufficiently large $L$, we apply Algorithm 1 twice using $\boldsymbol{T}^L$ to determine the next node to visit. In line 4, we choose the new node based on the order of $\pi_1$ and $\pi_2$, respectively. Once Algorithm 1 terminates, we obtain two vectors $\boldsymbol{h}$ and $\widehat{\boldsymbol{h}}$. We claim that $\boldsymbol{h} = \widehat{\boldsymbol{h}}$.

Suppose that we erase non-trivial positive cycles $C^1, \cdots, C^k$ in order when we choose the new node based on the order of $\pi_1$. If $k = 0$, then there is no need for erasing cycles, and in this case $\boldsymbol{h} = \widehat{\boldsymbol{h}}$. Now we consider $k > 0$, that is, there is at least one positive cycle to be erased. For $i = 1, \cdots, k$, we use $\boldsymbol{c}^i = (c_1^i, \cdots, c_n^i)^\top$ to denote the position of cycle $C^i$ in matrix $\boldsymbol{T}$. Then for $i \in \{1, \cdots, k-1\}$ and $j \in V$, we have

$$c_j^{i+1} = \begin{cases} 0 & \text{if } j \notin C^{i+1}, \\ \max\{c_j^1, \cdots, c_j^i\} + 1 & \text{if } j \in C^{i+1}. \end{cases} \tag{17}$$

Moreover, for $i \in V$, we have that $h_i = \max\{c_i^1, \cdots, c_i^k\} + 1$.

Now, suppose we choose the new node based on the order of $\pi_2$, and the first non-trivial positive cycle to be erased is $\widehat{C}^1$. Let $\widehat{\boldsymbol{c}}^1 = (\widehat{c}_1^1, \cdots, \widehat{c}_n^1)^\top$ denote the position of $\widehat{C}^1$ in matrix $\boldsymbol{T}^L$. For $i \in V$, either $\widehat{c}_i^1 = 0$ and node $i$ is not in cycle $\widehat{C}^1$, or $\widehat{c}_i^1 = 1$ and node $i$ belongs to cycle $\widehat{C}^1$. Since $\widehat{C}^1$ is a non-trivial positive cycle, there exists $i \in \widehat{C}^1$ such that $h_i > 1$. This implies that $\widehat{C}^1$ must have some common nodes with cycles $C^1, \cdots, C^k$ that have the same position in matrix $T$. Suppose $C^i$ is the first cycle that has some common nodes with $\widehat{C}^1$. If $\widehat{\boldsymbol{c}}^1 \neq \boldsymbol{c}^i$, then there is a common node $j \in \widehat{C}^1 \cap C^i$ such that $\widehat{c}_j^1 = 1 \neq c_j^i$. This implies that $c_j^i > 1$, which contradicts the fact that $C^i$ is the first cycle having some common nodes with $\widehat{C}^1$. Therefore, $\widehat{C}$ and $C^i$ must be the same cycle, and $\widehat{\boldsymbol{c}}^1 = \boldsymbol{c}^i$. In other words, $\widehat{C}^1 \in C^1, \cdots, C^k$.

Suppose we have erased non-trivial positive cycles $\widehat{C}^1, \ldots, \widehat{C}^u$ based on the order of $\pi_2$, and for $i = 1, \ldots, u$, we have $\widehat{C}^i \in C^1, \ldots, C^k$. If $u < k$, then the algorithm will not terminate and the next positive cycle to be erased is $\widehat{C}^{u+1}$. Following the previous proof, we can show that $\widehat{C}^{u+1} \in C^1, \ldots, C^k$. If $u = k$, then the algorithm terminates. Therefore, if we choose new nodes based on the order of $\pi_2$, only the order of positive loop-erasure will be changed, and we will still have $\boldsymbol{h} = \widehat{\boldsymbol{h}}$.

As a result, given a fixed matrix $\boldsymbol{T}^L$ with sufficiently large $L$, the random walk order will not affect the time complexity or the return result of Algorithm 1. Thus, if we randomly generate $T$, the expected time complexity of Algorithm 1 is independent of the random walk order. $\square$

### A.6. Proof of Theorem 5.2

**Proof.** The expected time complexity of Algorithm 1 can be expressed as the expected value of the $\ell_1$-norm of $h$ when performing Algorithm 1 over all possible matrices $T^L$. This can be written as $\mathbb{E}\left(\sum_{i=1}^n h_i\right) = \sum_{i=1}^n \mathbb{E}(h_i)$, where the equality follows from the linearity of the expectation. As shown in the proof of Lemma 5.1, the expected value of $h_i$, denoted by $\mathbb{E}(h_i)$, is independent of the order in which the random walk starts at each node.

Suppose that the random walk starts at node $v_1$. We can estimate $\mathbb{E}(h_1)$ as the expected number of times the walk visits node $v_1$ before terminating. Recall that termination occurs either when a negative cycle is encountered or when a root node is added to the branch at a node $u$, with probability $\frac{1}{1+d_u}$. We can derive an upper bound for $\mathbb{E}(h_1)$ by considering the case where the walk only stops when a root node is added, ignoring the possibility of stopping at negative cycles. In this case, the probability transition matrix is $P = (I + D)^{-1}(A^+ - A^-)$. The expected number of visits to node $v_1$ until termination can be calculated as $\lim_{t\to\infty} \sum_{i=1}^t e_1^\top (I + P + \cdots + P^t)e_1$. Since the walk in Algorithm 1 also terminates when encountering negative cycles, we have the following upper bound:

$$
\begin{aligned}
\mathbb{E}(h_1) &\leq \lim_{t\to\infty} e_1^\top (I + P + \cdots + P^t)e_1 \\
&= e_1^T (I + D - A^+ + A^-)^{-1}(I + D)e_1.
\end{aligned}
\tag{18}
$$

After summing the expected number of visits for all nodes, we obtain: $\sum_{i=1}^n \mathbb{E}(h_i) \leq \text{trace}((I+D-A^+ +A^-)^{-1}(I+D))$.

Let $\widehat{L}$ be the matrix $D - A^+ + A^-$, which is the Laplacian matrix of an unsigned directed graph $\widehat{\mathcal{G}} = (V, E, \widehat{w})$, where $\widehat{w}_{ij} = |w_{ij}| = 1$. The entry at row $i$ and column $j$ of $\widehat{L}$ is denoted by $l_{ij}$. We have $l_{ii} = d_i$ and $l_{ij} = 0$ or $-1$. Furthermore, the sum of all entries in each row of $\widehat{L}$ is equal to 0.

Matrix $\widehat{Q} = (I + \widehat{L})^{-1} = (\widehat{q}_{ij})_{n\times n}$ is the forest matrix on unsigned graph $\widehat{\mathcal{G}}$. From (Sun & Zhang, 2023), we have $\frac{1}{1+d_i} \leq \widehat{q}_{ii} \leq \frac{2}{2+d_i}$. Then we can derive the following inequality:

$$
\begin{aligned}
\sum_{i=1}^n \mathbb{E}(h_i) &\leq \text{trace}((I + D - A^+ + A^-)^{-1}(I + D)) \\
&= \sum_{i=1}^n \widehat{q}_{ii}(1 + d_i) \leq \sum_{i=1}^n \frac{2(1 + d_i)}{2 + d_i} \leq 2n.
\end{aligned}
\tag{19}
$$

As a result, the expected time complexity of algorithm 1 is at most $O(n)$. $\quad\square$

### A.7. Proof of Lemma 6.1

**Proof.** In Algorithm 1, suppose that the random walk is currently at node $i$, and a new step is needed. There are two possible scenarios: either node $i$ becomes a root node, or the walk moves from node $i$ to a random neighbor $j$. Both events occur with a probability of $\frac{1}{1+d_i}$. Consequently, the probability of obtaining any particular generalized spanning converging forest $\phi_0$ from $\mathcal{F}$ using Algorithm 1 is proportional to $\prod_{i=1}^n \frac{1}{1+d_i}$. Therefore, each forest $\phi_0 \in \mathcal{F}$ can be generated with equal likelihood, which completes the proof. $\quad\square$

### A.8. Proof of Lemma 6.2

**Proof.** With Lemma 6.1, we establish that for each $k = 1, \cdots, l$, the forest $\phi_k$ is uniformly sampled from $\mathcal{F}$. Consequently, the expected value of the estimator $\widehat{w}_l(\mathcal{F})$ is given by:

$$
\mathbb{E}(\widehat{w}_l(\mathcal{F})) = \mathbb{E}\left(\frac{|\mathcal{F}|}{l} \sum_{k=1}^l 2^{n^-(\phi_k)}\right) = \sum_{\phi\in\mathcal{F}} 2^{n^-(\phi)} = w(\mathcal{F}).
\tag{20}
$$

Similarly, the expected value of $\widehat{w}_l(\mathcal{F}_{ij})$ is calculated as follows:

$$
\begin{aligned}
\mathbb{E}(\widehat{w}_l(\mathcal{F}_{ij})) &= \mathbb{E}\big(\frac{|\mathcal{F}|}{l}\sum_{k=1}^{l}2^{n^-(\phi_k)}\mathrm{sign}(P_{ij})\mathbb{I}_{\{r_{\phi_k}(i)=j\}}\big) \\
&= \sum_{\phi\in\mathcal{F}}2^{n^-(\phi)}\mathrm{sign}(P_{ij})\mathbb{I}_{\{r_\phi(i)=j\}} \\
&= \sum_{\phi\in\mathcal{F}_{ij}}2^{n^-(\phi)}\mathrm{sign}(P_{ij}) = w(\mathcal{F}_{ij}).
\end{aligned}
\tag{21}
$$

This computation confirms that $\widehat{w}_l(\mathcal{F})$ and $\widehat{w}_l(\mathcal{F}_{ij})$ are indeed unbiased estimators for $w(\mathcal{F})$ and $w(\mathcal{F}_{ij})$, respectively, which completes the proof. $\square$

### A.9. Hoeffding's inequality

**Lemma A.1** (Hoeffding's inequality (Hoeffding, 1994)). *Let* $x_1, x_2, \cdots, x_l$ *be* $l$ *independent random variables satisfying* $a \le x_i \le b$ *for all* $i = 1, 2, \cdots, n$. *Let* $x = \frac{1}{l}\sum_{i=1}^{l}x_i$. *Then for any* $\epsilon > 0$, $\mathbb{P}(|x - \mathbb{E}(x)| \ge \epsilon) \le 2\exp\left(-\frac{2l\epsilon^2}{(b-a)^2}\right)$.

### A.10. Proof of Theorem 6.3

**Proof.** In a balanced signed graph $\mathcal{G}$, there are no negative cycles. Then we have $\widehat{q}_{ij} = \frac{1}{l}\sum_{k=1}^{l}\mathrm{sign}(P_{ij})\mathbb{I}_{\{r_{\phi_k}(i)=j\}}$, $\widehat{q}_{ii} = \frac{1}{l}\sum_{j=1}^{l}\mathbb{I}_{\{i\in\mathcal{R}(\phi_j)\}}$, $\widetilde{q}_{ii} = \frac{1}{1+d_i}(1 + \sum_{k\neq i}\widehat{q}_{ik}w_{ki})$. Since $\phi_1, \phi_2, \cdots, \phi_l$ are independently and uniformly sampled from the set $\mathcal{F}$, the sample size $l$ does not influence the relative variances $\widetilde{q}_{ii}$ and $\widehat{q}_{ii}$. For simplicity, we assume $l = 1$ for the remainder of this proof. Under this assumption, the variance of $\widehat{q}_{ii}$ is $\mathrm{Var}(\widehat{q}_{ii}) = q_{ii} - q_{ii}^2$. The variance of $\widetilde{q}_{ii}$ can be derived as follows:

$$
\begin{aligned}
\mathrm{Var}(\widetilde{q}_{ii}) &= \mathbb{E}(\widetilde{q}_{ii}^2) - (\mathbb{E}(\widetilde{q}_{ii}))^2 = \frac{1}{(1+d_i)^2}\mathbb{E}\big((1 + \sum_{k\neq i}\widehat{q}_{ik}w_{ki})^2\big) - q_{ii}^2 \\
&= \frac{1}{(1+d_i)^2}\mathbb{E}(1 + 2\sum_{k\neq i}\widehat{q}_{ik}w_{ki} + (\sum_{k\neq i}\widehat{q}_{ik}w_{ki})^2) - q_{ii}^2 \\
&= \frac{1 + 3\sum_{k\neq i}q_{ik}w_{ki}}{(1+d_i)^2} - q_{ii}^2 = \frac{1 + 3((1+d_i)q_{ii} - 1)}{(1+d_i)^2} - q_{ii}^2 \\
&= \frac{3q_{ii}}{1+d_i} - \frac{2}{(1+d_i)^2} - q_{ii}^2.
\end{aligned}
\tag{22}
$$

The simplification uses the assumptions that $\mathbb{E}(\widehat{q}_{ik}\widehat{q}_{is}) = 0$ for any $k \neq s \neq i$ and $\mathbb{E}(\widehat{q}_{ik}^2) = |q_{ik}| = q_{ik}w_{ki}$ in balanced signed graphs. Then we get the following equality:

$$
\mathrm{Var}\{\widehat{q}_{ii}\} - \mathrm{Var}\{\widetilde{q}_{ii}\} = \frac{2(1 - q_{ii})}{(1+d_i)^2} + \frac{d_i(d_i - 1)q_{ii}}{(1+d_i)^2} \ge 0.
\tag{23}
$$

It shows that the variance of $\widetilde{q}_{ii}$ is no more than the variance of the estimator $\widehat{q}_{ii}$, which completes the proof. In fact, Theorem 6.3 extends Lemma 6.1 from (Sun & Zhang, 2024), as the unsigned case can be viewed as a special case of balanced signed graphs. $\square$

### A.11. Proof of Theorem 6.4

**Proof.** Setting $a = 0$ and $b = |\mathcal{F}|\alpha$, and choosing $l$ as previously specified, we can prove the inequalities (2) and (3) directly by utilizing Hoeffding's inequality. Assuming the above inequalities hold, the error in the estimated ratio can be

bounded as follows:

$$|\widehat{\boldsymbol{q}}[i] - q_{ii}| = \left| \frac{\widehat{w}_l(\mathcal{F}_{ii})}{\widehat{w}_l(\mathcal{F})} - \frac{w(\mathcal{F}_{ii})}{w(\mathcal{F})} \right|$$

$$= \left| \frac{w(\mathcal{F}_{ii})(\widehat{w}_l(\mathcal{F}) - w(\mathcal{F})) + w(\mathcal{F})(w(\mathcal{F}_{ii}) - \widehat{w}_l(\mathcal{F}_{ii}))}{\widehat{w}_l(\mathcal{F})w(\mathcal{F})} \right|$$

$$\leq \frac{w(\mathcal{F}_{ii})|\widehat{w}_l(\mathcal{F}) - w(\mathcal{F})| + w(\mathcal{F})|w(\mathcal{F}_{ii}) - \widehat{w}_l(\mathcal{F}_{ii})|}{\widehat{w}_l(\mathcal{F})w(\mathcal{F})} \tag{24}$$

$$\leq \frac{\frac{\epsilon\beta}{2+\epsilon}(w(\mathcal{F}_{ii}) + w(\mathcal{F}))|\mathcal{F}|}{\widehat{w}_l(\mathcal{F})w(\mathcal{F})} \leq \frac{\frac{2\epsilon\beta}{2+\epsilon}}{\beta - \frac{\epsilon\beta}{2+\epsilon}} = \epsilon,$$

where the last inequality holds since $w(\mathcal{F}_{ii}) \leq w(\mathcal{F})$, $w(\mathcal{F}) = |\mathcal{F}|\beta$ and $\widehat{w}_l(\mathcal{F}) \geq w(\mathcal{F}) - |\mathcal{F}|\frac{\epsilon\beta}{2+\epsilon}$, which finishes the proof. $\square$

## B. Pseudocodes for Algorithms

### B.1. Pseudocode for Algorithm GSCF

---
**Algorithm 1** GSCF($\mathcal{G}$)

---
1: **Input:** Signed graph $\mathcal{G} = (V, E, w)$ with $|V| = n$
2: **Output:** Generalized spanning converging forest $\phi$
3: **Initialize:** $\phi \leftarrow \emptyset$; $V_\phi \leftarrow \emptyset$; $E_\phi \leftarrow \emptyset$
4: **for** $i = 1, 2, \cdots, n$ **do**
5:     $u \leftarrow i$
6:     Create a branch $P \leftarrow \emptyset$
7:     **while** $u \notin V_\phi$ **do**
8:         seed $\leftarrow$ RAND$(0, 1)$
9:         **if** seed $\leq \frac{1}{1+d_u}$ **then**
10:           Mark $u$ as the root node
11:           **break**
12:         **else**
13:           Select a random neighbor $v \in N(u)$
14:           Add edge $(u, v)$ to $P$
15:           **if** $P$ has a negative cycle $C$ **then**
16:             **break**
17:           **else**
18:             $u \leftarrow v$
19:           **end if**
20:         **end if**
21:     **end while**
22:     **if** $P$ has a negative cycle $C$ **then**
23:         Partition $P$ into $P'$ and $C$
24:         Perform loop-erasure on $P'$ and obtain $P'_{\text{LE}}$
25:         Add $P'_{\text{LE}}$ and $C$ to $\phi$; update $V_\phi$ and $E_\phi$
26:     **else**
27:         Perform loop-erasure on $P$ and obtain $P_{\text{LE}}$
28:         Add $P_{\text{LE}}$ to $\phi$; update $V_\phi$ and $E_\phi$
29:     **end if**
30: **end for**
31: **return** $\phi$

---

## B.2. Pseudocode for Algorithm FMDE/FMDE+

---

**Algorithm 2** FMDE/FMDE+$(\mathcal{G}, l)$

---

1: **Input:** Signed graph $\mathcal{G}$; sample number $l$
2: **Output:** $\widehat{q}$ (FMDE estimator), $\widetilde{q}$ (FMDE+ estimator)
3: **Initialize:** $\widehat{q}[i] \leftarrow 0$, $\widetilde{q}[i] \leftarrow 0$ for $i = 1, \ldots, n$; $\gamma \leftarrow 0$
4: **for** $t = 1, 2, \ldots, l$ **do**
5: $\quad \phi \leftarrow \text{GSCF}(\mathcal{G})$
6: $\quad \gamma \leftarrow \gamma + 2^{n^-(\phi)}$
7: $\quad$ **for** $i = 1, 2, \ldots, n$ **do**
8: $\quad\quad j \leftarrow r_\phi(i)$
9: $\quad\quad$ **if** $j = i$ **then**
10: $\quad\quad\quad \widehat{q}[i] \leftarrow \widehat{q}[i] + 2^{n^-(\phi)}$
11: $\quad\quad$ **end if**
12: $\quad\quad$ **if** $j > 0$ **and** $i \in N(j)$ **then**
13: $\quad\quad\quad \widetilde{q}[i] \leftarrow \widetilde{q}[i] + \text{sign}(P_{ij}) \, w_{ji} \, 2^{n^-(\phi)}$
14: $\quad\quad$ **end if**
15: $\quad$ **end for**
16: **end for**
17: $\widehat{q} \leftarrow \widehat{q}/\gamma$
18: **for** $i = 1, 2, \ldots, n$ **do**
19: $\quad \widetilde{q}[i] \leftarrow \frac{\widetilde{q}[i]}{\gamma(1+d_i)} + \frac{1}{1+d_i}$
20: **end for**
21: **return** $\widehat{q}, \widetilde{q}$

---

## B.3. Pseudocode for Algorithm FJOE

---

**Algorithm 3** FJOE$(L, i, \boldsymbol{s})$

---

1: **Input:** List $L$ of $l$ generalized spanning converging forests; node index $i$; internal opinion vector $\boldsymbol{s}$
2: **Output:** Estimated expressed opinion $\widehat{z_i}$ for node $i$
3: **Initialize:** $\widehat{z_i} \leftarrow 0$; $\gamma \leftarrow 0$
4: **for all** $\phi \in L$ **do**
5: $\quad \eta \leftarrow 2^{n^-(\phi)}$
6: $\quad \gamma \leftarrow \gamma + \eta$
7: $\quad k \leftarrow r_\phi(i)$
8: $\quad$ **if** $k \neq 0$ **then**
9: $\quad\quad \widehat{z_i} \leftarrow \widehat{z_i} + \text{sign}(P_{ik}) \eta \, s_k$
10: $\quad$ **end if**
11: **end for**
12: $\widehat{z_i} \leftarrow \widehat{z_i}/\gamma$
13: **return** $\widehat{z_i}$

---

# C. Datasets and Equipment

## C.1. Equipment and Implementation Details

All experiments are conducted using the Julia programming language in a computational environment equipped with a 2.10 GHz Intel(R) Xeon(R) Platinum 8352V CPU and 256GB of primary memory. For all algorithms, the number of generalized spanning converging forests $l$ is set according to Theorem 6.4, with parameters $\delta = 0.01$. Since real networks usually contain very few negative edges, and in balanced signed graphs $\alpha/\beta = 1$. Since $\alpha$ and $\beta$ are difficult to compute exactly, we set $\alpha/\beta = 2$ as a conservative choice. Moreover, the bound in Theorem 6.4 is loose in practice; e.g., as shown in Figure 2, when $\epsilon = 0.3$, FMDE+ achieves an average relative error of about 0.01. Given that our sampling algorithms can be

parallelized efficiently, we use 72 computing cores to speed up the process.

## C.2. Datasets

The datasets of selected real networks are publicly available in the KONECT (Kunegis, 2013) and SNAP (Leskovec & Sosič, 2016). Our experiments are conducted on a diverse range of networks, with node counts ranging from 2,539 to over 23 million and edge counts from 12,969 to 112 million. Details of these datasets are presented in Table 3, which includes six small graphs along with six medium and large-sized graphs. We utilize both original signed graphs and modified signed graphs for our experiments. These modified signed graphs are denoted with a superscript asterisk in Table 3.

*Table 3.* Datasets used in experiments.

| Type | Network | Nodes | Edges |
|---|---|---|---|
| Small Graphs | Adolescent* | 2,539 | 12,969 |
| | Bitcoinotc | 5,881 | 35,592 |
| | Gnutella08* | 6,301 | 20,777 |
| | Wikielec | 7,118 | 103,675 |
| | Wikipedia* | 17,649 | 296,918 |
| | SlashdotZoo | 79,120 | 515,397 |
| Medium and Large Graphs | Epinions | 131,828 | 841,372 |
| | WikiL | 258,259 | 3,187,096 |
| | Youtube* | 1,134,890 | 2,987,624 |
| | Dblp* | 5,624,219 | 12,282,055 |
| | Livejournal* | 7,489,073 | 112,307,315 |
| | FullUSA* | 23,947,300 | 57,708,600 |

