# OpenReview forum: "Fast Estimation for Forest Matrix of Signed Graphs"
_ICML.cc/2026/Conference — ICML 2026 regular_

### Official Review · Reviewer_mCxR · 2026-03-12

**Soundness:** 2
**Presentation:** 3
**Significance:** 3
**Originality:** 3
**Overall Recommendation:** 4
**Confidence:** 5

**Summary:**

This paper generalizes the matrix-forest theorem to signed graphs and estimates the signed forest matrix efficiently, then introduces a loop-erased–random-walk sampler that generates generalized forests on signed graphs, and a variance-reduced method which reduces the estimation error by about an order of magnitude for the diagonal. Finally, this paper apply the aforementioned methods to signed Friedkin–Johnsen opinion dynamics, where the approach works very well in practice and still comes with theoretical guarantees.

**Compliance With Llm Reviewing Policy:**

Affirmed.

**Key Questions For Authors:**

1. Since balanced signed graphs can be switched so that unsigned methods become applicable, can prior unsigned approaches be directly adapted for the balanced case? If so, what is the main advantage of your method in this regime?
2. Can you provide a more complete and convincing proof  of Lemma 6.1 establishing that GSCF samples generalized spanning converging forests uniformly?
3. Can you explicitly discuss the limitations of your method？

**Limitations:**

yes

**Strengths And Weaknesses:**

S1. This paper proposes a novel signed-graph matrix–forest theorem with a matching random-walk sampler, and give a theoretically justified variance-reduction improvement.

S2. The experiments clearly demonstrate the effectiveness of FMDE and variance-reduction benefits from FMDE+. FMDE+ is consistently more accurate and run efficiently , while on large graphs where EXACT is computationally infeasible, and the signed opinion-dynamics study likewise scales to large instances with fast runtimes.

S3. The paper is clearly written and well motivated, with a smooth thread from theory to algorithms to applications and some accessible examples. The work extends the matrix–forest theorem to signed graphs and provides a practical estimator that scales to large graphs.

W1. The uniform-sampling claim in Lemma 6.1 is not convincingly justified. The argument effectively treats the return probability as depending only on $\prod_i \frac{1}{1+d_i}$, seemingly ignoring path-dependent termination and cycle-erasure interactions, even though this uniformity is central to the unbiasedness of later estimators. In Lemma 4.4, there is a sign/algebra error in the step $\sum_{k \ne i} |q_{ik}|$ being replaced by $q_{ii}$ (leading to $\frac{1}{1+d_i}(1+q_{ii})$), whereas it should relate to $1-q_{ii}$.

W2. The experimental section reports runtime/scalability but does not provide space or memory footprint measurements.

W3. In Section 7.2.1 (ACCURACY), there is an unresolved cross-reference “Figure??” that should be corrected to the proper figure number.

W4. The paper lacks a dedicated limitations/impact discussion.

---

> ### Author Rebuttal · Authors · 2026-03-30
>
> Thank you for your time and helpful comments. Below is a summary of our response to your concerns.
>
> **Response to W1 and Q2.**
>
> GSCF returns a generalized spanning converging forest $\phi$. In the resulting forest, each node has either out-degree $1$, or out-degree $0$ if it is a root. For each non-root node $i$, selecting its outgoing neighbor occurs with probability $\frac{1}{1 + d_i}$; for each root node $i$, stopping at itself to become a root also has probability $\frac{1}{1 + d_i}$. Therefore, the probability of generating a particular forest $\phi$ is proportional to $\prod_{i} \frac{1}{1 + d_i}$, which is identical for all generalized spanning converging forests $\phi$.
>
> Moreover, although the execution of Algorithm 1 depends on the realized random walk paths, the proof of Lemma 5.1 shows that the distribution of the final output is independent of the order in which the random walks are initiated from different nodes. In addition, since positive cycles are always erased and do not affect the final structure, they do not influence the sampling distribution. Therefore, Algorithm 1 achieves uniform sampling over all generalized spanning converging forests. We will revise this part to make the argument more explicit, rigorous, and easier to understand.
>
> In Lemma 4.4, we will correct $q_{ii}$ to $1 - q_{ii}$. Thank you for your careful reading.
>
> **Response to W2 and W3.**
>
> The space complexity of our algorithm is linear $O(n)$. The memory overhead mainly comes from storing arrays used during random walks, which is on the same order as the graph size. We will include a more detailed discussion of memory usage in the revised version.
>
> The figure reference in Line 358 should be Figure 2. We will correct this in the revised version.
>
> **Response to W4 and Q3.**
>
> Thank you for your helpful suggestions.
>
> One limitation of our method is that both efficiency and accuracy may decrease when the graph contains a large number of negative cycles. However, prior studies (e.g., Chiang et al., 2014; Singh and Adhikari, 2017) suggest that real-world signed networks typically contain relatively few negative cycles. In practice, our method still performs well on real-world datasets.
>
> Another limitation is that, although we have evaluated our method on the signed FJ model, we have not yet explored other applications such as signed link prediction. We consider this an important direction for future work.
>
> We will add a dedicated discussion of these limitations in the revised version.
>
> **Response to Q1.**
>
> Even for balanced signed graphs, negative edges still exist, and thus classical matrix-forest theorems and sampling methods for unsigned graphs are not directly applicable. In this work, we establish a matrix-forest theorem for signed graphs and develop corresponding sampling algorithms with complexity analysis, which apply to both balanced and unbalanced signed graphs.
>
> Our analysis also reveals that, in balanced graphs, certain results exhibit similarities to the unsigned case. However, real-world signed networks are often unbalanced, and all datasets used in our experiments are unbalanced signed graphs.
>
> Thank you again for your insightful and helpful suggestions.

---

> > ### Author Rebuttal · Reviewer_mCxR · 2026-04-03
> >
> > Thank you for the response. It addresses my main concerns, so I will keep my score unchanged.

---

### Official Review · Reviewer_2ic4 · 2026-03-12

**Soundness:** 2
**Presentation:** 2
**Significance:** 2
**Originality:** 3
**Overall Recommendation:** 4
**Confidence:** 3

**Summary:**

The key contribution of this work is the efficient estimation of the forest matrix for signed graphs, particularly the diagonal entries of $Q = (I + L)^{-1}$, where $L$ is the signed Laplacian matrix. Specifically, the paper makes the following contributions:

1. Signed Forest Matrix Theorem: An extension of the classical matrix-forest theorem to signed graphs using generalised spanning converging forests.

2. GSCF algorithm: A variant of loop-erased random walks to sample generalised spanning converging forests with expected $O(n)$ runtime.

3. FMDE and FMDE+ algorithms: Sampling-based estimators for the diagonal of the forest matrix with time complexity $O(ln)$, where $l$ is the number of samples.

4. FJOE algorithm: A method to estimate the expressed opinion of the signed Friedkin–Johnsen model.

**Compliance With Llm Reviewing Policy:**

Affirmed.

**Final Justification:**

I thank the authors for their response and it addresses my concerns. I will be increasing my score to reflect this.

**Key Questions For Authors:**

1. In the classical matrix-forest theorems (Chebotarev and Shamis), the weight of a subgraph with at least one edge is defined as the product of the weight of all the edges. The weight of a subgraph without edges is assumed to be $1$.

   In this manuscript, the weight of a generalised spanning converging forest $\phi$ is defined as $w(\phi) = 2^{n^-(\phi)}$, where $n^-(\phi)$ denotes the number of non-trivial negative cycles in $\phi$. If there is no edge in $\phi$, its weight is defined to be $1$.

   What does $\mathrm{det}(I + L)$ represent in both cases?

2. In Line $116$ second column rooted convergence tree is not defined in the manuscript.

3. In Line $125$, the function $r_\phi$ for each node $i$ is defined. It is not clear why $r_\phi$ is a function?

4. How does the algorithm GSFC modify to work on negative cycles? How different is the analysis from (Wilson, 1996)?

5. Is the method limited to diagonals? Is it possible to estimate non-diagonal entries efficiently?

**Limitations:**

This work only formulates the forest matrix theorem for the signed graphs. Can it be done for weighted directed graphs.

**Strengths And Weaknesses:**

Strengths:

1. The signed forest matrix theorem formulates the relation between the forest matrix $Q = (I + L)^{-1}$ of signed graphs and the generalised spanning converging forest, which is a new theoretical contribution.

2. A modified loop-erased random walk method has been proposed to generate a generalised spanning converging forests as negative cycles are allowed.

3. The proposed methods scale to very large graphs (millions of nodes).

Weaknesses:

1. The practical implication of the forest matrix of a signed graph in core machine learning problems is weak.

2. The manuscript lacks motivation on why generalised spanning converging forests are a natural extension of the forest matrix theorem for signed graphs.

3. Some results depend on balanced signed graphs.

---

> ### Author Rebuttal · Authors · 2026-03-30
>
> Thank you for your time and helpful comments. Below is a summary of our response to your concerns.
>
> **Response to W1.**
>
> In recent years, the forest matrix and its variants have been applied across a wide range of areas, including opinion dynamics (Gionis et al., 2013; Sun and Zhang, 2023;  Xu et al., 2021; Neumann et al., 2024; Sun et al., 2025), graph signal processing (Pilavcı et al., 2021, 2020), and Markov processes (Avena and Gaudillière, 2018; Avena et al., 2018) , all of which are closely related to modern AI and machine learning.
>
> Moreover, the forest matrix can be viewed as a diffusion matrix closely related to the PageRank matrix, which is widely used in node ranking and prediction tasks in machine learning [1,2]. In addition, several recent works have explored diffusion- or propagation-based formulations, which are closely related to the forest matrix, improve the efficiency and scalability of graph neural network computations [3,4]. We will include these references and clarify this connection in the revised version.
>
> In this work, we focus on the efficient computation of the forest matrix in signed graphs and its application to the signed FJ model. We will consider extending our framework to other applications, such as signed link prediction and related learning problems, in future work.
>
> [1] Page et al. The PageRank Citation Ranking: Bringing Order to the Web. 1999.
>
> [2] Andersen et al. Local Graph Partitioning Using PageRank Vectors. FOCS 2006.
>
> [3] Klicpera et al. Predict then Propagate: Graph Neural Networks Meet Personalized PageRank. ICLR 2019.
>
> [4] Bojchevski et al. Scaling Graph Neural Networks with Approximate Personalized PageRank. KDD 2020.
>
> **Response to W2, W3, and Q1.**
>
> Thank you for these helpful comments. Our motivation comes from the signed extension of the matrix-forest theorem. When deriving the result for signed graphs, we find that negative cycles play a fundamental role, which correspond exactly to generalized spanning converging forests. Therefore, generalized spanning converging forests arise naturally as the signed extension of rooted forests in the classical matrix-forest theorem.
>
> In an unweighted unsigned graph, $\det(I+L)$ equals the number of rooted forests. In a signed graph with edge weights in $\{\pm1\}$, $\det(I+L)$ equals the total weight of all generalized spanning converging forests, where each forest has weight $2^k$ and $k$ is the number of negative cycles. For example, in Figure 1, $I+L = [3\ -1\ 1;\ 0\ 2\ 1;\ -1\ 0\ 2]$ and $\det(I+L)=15$. Here, $\phi_1$-$\phi_9$ have no negative cycle (weight $1$), while the remaining three each contain one negative cycle (weight $2$), giving $9+3\times2=15$. When all edges in Figure 1 are positive, $I+L = [3\ -1\ -1;\ 0\ 2\ -1;\ -1\ 0\ 2]$ and $\det(I+L)=9$. In this case, only $\phi_1$-$\phi_9$ remain, verifying that in the unsigned case $\det(I+L)$ equals the number of rooted forests.
>
> Our theorem, algorithms, and complexity analysis hold for unbalanced signed graphs. Only a small part assumes balanced graphs for technical convenience. All datasets used in our experiments are unbalanced signed graphs.
>
> We will revise the manuscript to clarify these points.
>
> **Response to Q2 and Q3.**
>
> A rooted converging tree is a directed tree in which all nodes have a unique path pointing toward a node called the root. Each node has out-degree either $1$ (pointing to its successor) or $0$ (if it is the root).
>
> Regarding the definition of $r_{\phi}(i)$, it is indeed a function. By definition, a generalized spanning converging forest $\phi$ consists of a collection of trees together with possible negative cycles. If a node $i$ lies on a negative cycle, we define $r_{\phi}(i)=0$. Otherwise, if $i$ is not on a negative cycle, we follow its outgoing edge recursively. If $i$ is a root, then $r_{\phi}(i)=i$; if not, we continue along its outgoing edges until reaching the root. Therefore, for nodes not in negative cycles, $r_{\phi}(i)$ returns the root of the connected component containing $i$. Thus, $r_{\phi}: V \to V \cup$ {0} is a well-defined function.
>
> We will revise the manuscript to clarify these definitions.
>
> **Response to Q4 and Q5.**
>
> Compared with Wilson’s algorithm, which erases all cycles to generate spanning trees, GSFC only erases positive cycles while retaining negative cycles, thus generating generalized spanning converging forests. Our method extends Wilson’s algorithm not only from spanning trees to forests, but more importantly by preserving negative cycles. This is necessary since the signed matrix-forest theorem depends on signed cycles, and negative cycles contribute to the sampling weights.
>
> Our method is not limited to diagonal entries; it naturally extends to non-diagonal entries. In the signed FJ model, the expressed opinion already involves both diagonal and off-diagonal entries (Line 322), and our method performs well in practice (Table 2).
>
> Thank you again for your insightful and helpful suggestions.

---

> > ### Author Rebuttal · Reviewer_2ic4 · 2026-04-04
> >
> > Follow-up on Q1:  What about the weighted multidigraph? How does Lemma 4.1 differ from the Matrix Forest Theorem for weighted multidigraphs (Theorem 3 in [1])? Since a signed graph is a special case of a weighted multidigraph, it would be helpful to clarify whether the proposed result is fundamentally new or follows as a direct consequence of existing results in the more general setting.
> >
> > [1] Chebotarev, P. Y. and Shamis, E. Matrix-forest theorems. ArXiv, abs/math/0602575, 200

---

> > > ### Author Response · Authors · 2026-04-04
> > >
> > > Thank you for this insightful question. Our framework can be naturally extended to the weighted directed graph (and multidigraph) settings. In the unweighted signed case, we define the weight of a generalized spanning converging forest $\phi$ as $w(\phi) = 2^{n^{-}(\phi)}$, where $n^{-}(\phi)$ denotes the number of non-trivial negative cycles. In the weighted case, this can be generalized to $w(\phi) = 2^{n^{-}(\phi)} \cdot \prod_{e \in \phi} |w_e|$, that is, we only need to additionally multiply the absolute values of edge weights. Under this definition, $\det(I+L)$ still equals the total weight of all generalized spanning converging forests.
> > >
> > > To illustrate, consider Figure 1 and modify the edge weights as follows: the edge from node 1 to node 3 has weight $-2$, and the edge from node 1 to node 2 has weight $3$, while the other edges remain unchanged. Then $I + L = [6\ -3\ 2;\ 0\ 2\ 1;\ -1\ 0\ 2]$ and $\det(I+L) = 31$. The corresponding generalized spanning converging forests $\phi_1$ to $\phi_{12}$ have weights $1, 1, 3, 3, 1, 3, 1, 2, 2, 6, 4, 4$, whose sum is also $31$, verifying the consistency.
> > >
> > > The multidigraph case can be handled in the same way by defining $L_{ij}$ as the negative sum of weights of all edges from $i$ to $j$. With this definition, the determinant interpretation extends directly. In our paper, we focus on unweighted graphs for notational simplicity, while our proofs and algorithms can be extended to weighted multidigraphs in a straightforward manner.
> > >
> > > Our main novelty lies in identifying the critical role of negative cycles in signed graphs and extending rooted forests to generalized spanning converging forests, together with a new weighting scheme that incorporates the factor $2^{n^{-}(\phi)}$. From this perspective, both our definition and theoretical formulation are new.
> > >
> > > On the other hand, by carefully defining generalized spanning converging forests and their weights, we ensure that $\det(I+L)$ still equals the total weight of all such forests. Therefore, our result is consistent in form with Theorem 3 in [1], and can be viewed as an extension of the matrix-forest theorem to the signed setting.
> > >
> > > Thank you again for your insightful question, and we hope this response has addressed your concern.

---

### Official Review · Reviewer_rMvh · 2026-03-13

**Soundness:** 3
**Presentation:** 3
**Significance:** 3
**Originality:** 4
**Overall Recommendation:** 5
**Confidence:** 4

**Summary:**

This paper studies fast estimation of the forest matrix for signed graphs. The main challenge is that classical forest-matrix results and sampling techniques for unsigned graphs do not directly extend to the signed setting because negative edges change the underlying combinatorial and algebraic structure. To address this, the paper introduces a signed-graph analogue of the forest matrix theorem based on generalized spanning converging forests, designs a corresponding sampling algorithm, and develops estimators for diagonal entries of the forest matrix. The paper also applies these estimators to signed Friedkin–Johnsen opinion dynamics, yielding a fast method for estimating expressed opinions. Empirically, the proposed methods show good scalability and substantially lower running time than exact matrix inversion while maintaining reasonable accuracy.

**Compliance With Llm Reviewing Policy:**

Affirmed.

**Final Justification:**

The authors' rebuttal has addressed my original questions. Therefore I recommend acceptance.

**Key Questions For Authors:**

Can the authors provide a more explicit explanation for why Algorithm 1 yields uniform sampling over generalized spanning converging forests, despite its seemingly path-dependent execution?

In the proof of Theorem 6.4, the final inequality appears to require a lower bound on $\hat w_l(F)$; is the displayed inequality direction there a typo?

How does the estimation error change as the sampling budget increases, and how closely does the observed behavior match the theoretical guarantees?

How sensitive are FMDE and FMDE+ to graph properties such as negative-edge ratio or degree heterogeneity?

There appears to be an unresolved figure reference in the experimental section; could the authors clarify and correct that reference?

I may raise my score if these concerns are addressed.

**Limitations:**

The empirical evaluation is somewhat limited in analyzing how accuracy varies with the sampling budget and with graph properties such as balance or negative-edge ratio.

The presentation of some theoretical arguments could be clearer, and there appears to be a minor typographical issue in the proof of Theorem 6.4.

**Strengths And Weaknesses:**

**Strengths**

The paper studies a meaningful and underexplored problem: fast estimation of the forest matrix in signed graphs, where standard techniques for unsigned graphs do not directly apply. The motivation is clear, and the paper explains well why the signed setting requires new ideas.

The paper proposes a coherent technical pipeline, from a signed-graph forest-matrix interpretation to a sampling procedure and then to estimators for diagonal entries and signed FJ opinion estimation. This gives the work a clear theorem-to-algorithm-to-application structure.

The proposed methods appear practically scalable. The experiments show substantial runtime advantages over exact matrix inversion, and the signed Friedkin–Johnsen application helps illustrate the practical relevance of the framework.

**Weaknesses**

The paper would benefit from a clearer presentation of some key theoretical arguments and assumptions. For example, the proof of Lemma 6.1 looks plausible, but it would be helpful to better explain why the seemingly complicated Algorithm 1 can be analyzed through such a simple argument.

While the experiments demonstrate scalability and promising accuracy, the empirical evaluation could be more comprehensive in showing how performance depends on graph structure and sampling budget.

---

> ### Author Rebuttal · Authors · 2026-03-30
>
> Thank you for your time and helpful comments. Below is a summary of our response to your concerns.
>
> **Explanation for Algorithm 1 and Lemma 6.1**:
>
> Although Algorithm 1 may appear complicated at first glance, its underlying logic is in fact quite simple. Algorithm 1 only erases positive cycles and ultimately returns a generalized spanning converging forest $\phi$. In the resulting forest, each node has either out-degree $1$, or out-degree $0$ if it is a root. For each non-root node $i$, selecting its outgoing neighbor occurs with probability $\frac{1}{1 + d_i}$; for each root node $i$, stopping at itself to become a root also has probability $\frac{1}{1 + d_i}$. Therefore, the probability of generating a particular forest $\phi$ is proportional to $\prod_{i} \frac{1}{1 + d_i}$, which is the same for all generalized spanning converging forests $\phi$. Hence, all such forests are sampled with equal probability. Moreover, although the execution of Algorithm 1 depends on the realized random walk paths, the proof of Lemma 5.1 shows that the distribution of the final output is independent of the order in which the random walks are initiated from different nodes. Therefore, Algorithm 1 achieves uniform sampling over all generalized spanning converging forests. We will revise this part to make the argument more explicit, rigorous, and easier to understand.
>
> **The Proof of Theorem 6.4 and unresolved figure reference.**
>
> Thank you for your careful reading. In Theorem 6.4, there is a typo in the last inequality (Line 836), where the direction of the inequality should be reversed. In addition, the figure reference in Line 358 should be Figure 2. We will correct both issues in the revised version.
>
> **Estimation error and sensitivity analysis.**
>
> As shown in Figure 2, the estimation error consistently decreases as the sampling budget increases. Moreover, we observe that the empirical errors are significantly smaller than the theoretical bounds, indicating that our analysis is relatively conservative. In particular, for FMDE+, with a sampling parameter corresponding to a theoretical error level of $0.3$, the actual observed error is around $0.01$ across all six datasets in Figure 2.
>
> Regarding sensitivity to graph properties, both our theoretical analysis and experimental results suggest that the running time and accuracy are not highly sensitive to degree heterogeneity. However, the number of negative cycles in the graph has a noticeable impact on sampling efficiency: graphs with fewer negative cycles tend to achieve lower estimation error under the same sampling budget. Fortunately, prior studies (e.g., Chiang et al., 2014; Singh and Adhikari, 2017) indicate that real-world signed networks typically contain relatively few negative cycles.
>
> Thank you again for your nice suggestions and we will include a more detailed discussion of these aspects in the revised version.

---

> > ### Author Rebuttal · Reviewer_rMvh · 2026-04-03
> >
> > The authors have effectively addressed my concerns in the rebuttal. Therefore, I am raising my score.

---

### Decision · Program_Chairs · 2026-04-30

**Decision:**

Accept (regular)

**Comment:**

The paper considers a natural problem for network science, proves new results on estimating its object of study -- namely, $(I+L)^{-1}$, where $L$ is the Laplacian matrix of the graph -- and contains a small empirical evaluation of the proposed methods. After the discussion, the reviewers were mostly on the "weak accept" range. I, on the other hand, think that this would be a fine ICML paper. On the other hand, one may argue that the paper is not on a core ML subject; as such, I would not mind it being bumped down in favor of more central contributions.